# Design of a Cereblon construct for crystallographic and biophysical studies of protein degraders

Alena Kroupova [1], Valentina A. Spiteri[1], Zoe J. Rutter[1], Hirotake Furihata[1], Darren Darren [1,2], Sarath Ramachandran [1,3], Sohini Chakraborti [1], Kevin Haubrich [1], Julie Pethe[1,4], Denzel Gonzales[1,5], Andre J. Wijaya[1,6], Maria Rodriguez-Rios [1], Manon Sturbaut [1], Dylan M. Lynch [1], William Farnaby [1], Mark A. Nakasone [1], David Zollman [1] ✉ & Alessio Ciulli [1] ✉

The ubiquitin E3 ligase cereblon (CRBN) is the target of therapeutic drugs thalidomide and lenalidomide and is recruited by most targeted protein degraders (PROTACs and molecular glues) in clinical development. Biophysical and structural investigation of CRBN has been limited by current constructs that either require co-expression with the adaptor DDB1 or inadequately represent full-length protein, with high-resolution structures of degrader ternary complexes remaining rare. We present the design of CRBN$^{midi}$, a construct that readily expresses from *E. coli* with high yields as soluble, stable protein without DDB1. We benchmark CRBN$^{midi}$ for wild-type functionality through a suite of biophysical techniques and solve high-resolution co-crystal structures of its binary and ternary complexes with degraders. We qualify CRBN$^{midi}$ as an enabling tool to accelerate structure-based discovery of the next generation of CRBN based therapeutics.

Targeted protein degradation enables development of small molecule therapies that promise to address medical needs that are currently unmet via conventional drugs[1,2]. Small-molecule degraders co-opt the ubiquitin-proteasome system by recruiting a specific target protein to a ubiquitin E3 ligase complex, thereby inducing protein ubiquitination and subsequent degradation. This is achieved through bifunctional proteolysis-targeting chimeras (PROTACs) that are composed of two distinct portions that separately bind the target and E3 ligase, or so-called "molecular glues" that are typically smaller than PROTACs and bind preferentially or exclusively either the E3 ligase or the target[3,4]. To date, over 40 PROTACs and molecular glues are under development in clinical trials for various

disease conditions through targeting a range of proteins for degradation[5,6].

Most of the current clinical candidates, and indeed many of the degraders reported in the literature to date, co-opt the protein cereblon (CRBN), the substrate receptor subunit of the Cullin4 E3 ligase complex (CRL4$^{CRBN}$). CRBN was identified in 2010 as the target of the infamous drug thalidomide, and related immunomodulatory drugs (IMiDs) such as lenalidomide that is approved for the treatment of multiple myeloma and other haematological malignancies[7,8]. Thalidomide and lenalidomide bind non-covalently to CRBN with structurally defined binding modes that mimic the C-terminal cyclic imide degron of CRBN substrates[9–11]. Upon binding to CRBN, IMiDs act as molecular

[1]Centre for Targeted Protein Degradation, School of Life Sciences, University of Dundee, Dundee DD1 5JJ, UK. [2]Present address: Cancer Science Institute Singapore, National University of Singapore, Singapore 117599, Singapore. [3]Present address: Biocon BMS R&D Center, Bommasandra Industrial Area, Bommasandra, Karnataka 560099, India. [4]Present address: National Heart and Lung Institute, Imperial College London, London SW3 6LY, UK. [5]Present address: Institute of Cell Biology, University of Edinburgh, Edinburgh EH9 3BF, UK. [6]Present address: PT Kalbe Farma, Jl. Let. Jend Suprapto Kav 4, Kalbe Farma, Jakarta 10510, Indonesia. ✉e-mail: d.l.zollman@dundee.ac.uk; a.ciulli@dundee.ac.uk

glues, enhancing the recruitment of a wide range of zinc-finger containing transcription factors and other proteins as CRBN *neo*-substrates leading to their rapid ubiquitination and degradation[12–14]. The degradation activity of IMiDs underpins their potent pleiotropic activity and explained thalidomide teratogenicity and lenalidomide anti-tumour efficacy. CRBN binders, including IMiDs and related chemotypes such as phenyl-glutarimides and dihydrouracil-containing compounds provide attractive drug-like starting points for the development of glue-type CRBN modulators and CRBN-recruiting PROTACs[15–17].

Despite the progress to date, the development of CRBN-based molecular glue and PROTAC degraders has witnessed limited enablement through structure-guided drug design. Indeed, few high-resolution structures of ternary complexes (CRBN:degrader:target) have been solved using either X-ray crystallography or cryo-EM[18–26]. Similarly, in vitro biophysical characterization of CRBN-based degraders has remained sparse[26–29]. This is in striking contrast with the structural and biophysical enablement that supports the design of PROTAC degraders recruiting another popular E3 ligase, von Hippel Lindau (VHL), as illustrated by the many ternary complex co-crystal structures solved by us and others that have led to efficient structure-guided design of VHL-based PROTACs[30–36], and their biophysical characterization[30,37,38]. Ternary co-crystal structures have also enabled structure-driven design of Cyclin K degraders which glue CDK12 to the CRL4 adaptor protein DDB1 (Damage specific DNA binding protein 1) to induce ubiquitination and degradation of CDK12-complexed Cyclin K[39–42].

In this work we design a truncated CRBN construct, CRBN[midi], and validate it using a range of biophysical techniques. We show that CRBN[midi] enables determination of high-resolution crystal structures of binary and ternary complexes with various ligands and degraders and their neo-substrate targets. In addition, we validate CRBN[midi] for use in binary and ternary complex formation assays using a range of biophysical techniques, including SPR, TR-FRET, ITC, nanoDSF, or SAXS. In all, we present CRBN[midi] as a tool that enables high-throughput structure-guided drug design of CRBN-based ligands and degraders.

## Results and Discussion

### Design of CRBN[midi]: a truncated CRBN construct containing Lon and TBD domains

We hypothesized that a major gap in structural enablement of CRBN has been largely caused by the challenges in producing and handling a suitable recombinant CRBN protein construct. CRBN consists of an unstructured N-terminal region and three folded domains: the Lon protease-like domain (Lon), the helical bundle (HB) which facilitates binding to the adaptor protein DDB1, and the thalidomide binding domain (TBD) (Fig. 1a). Upon ligand binding, the TBD and Lon domains undergo a ~45° rearrangement between an apo open and ligand-bound closed state[9,21,25]. Recombinant CRBN is insoluble when expressed in isolation but can be purified in complex with DDB1 from insect cells[9]. Whilst suitable for in vitro biochemical characterization, this full-length CRBN:DDB1 construct has not been crystallized so far. Several approaches have been developed to enable crystallization, including (i) a chimeric complex of *G. gallus* CRBN orthologue with human DDB1 (CRBN[G.gallus]:DDB1[H.sapiens])[9]; (ii) a construct lacking the unstructured N-terminal region of CRBN (CRBN[ΔN]:DDB1)[18,22,23,43]; and (iii) DDB1 lacking the WD40 propeller B (CRBN[ΔN]:DDB1[ΔBPB])[19–21,24,26]. The majority of the crystal structures that have resulted from these constructs have been solved to a resolution lower than 3 Å. Similarly, the full-length CRBN:DDB1 complex has been recently shown to be amenable to cryo-EM studies, however with overall resolution below 3 Å and often poorly resolved maps of the CRBN ligand binding site[25]. Alternatively, the isolated TBD of CRBN, spanning residues 328-426 (CRBN[TBD]), has been widely used because it is stable in solution and

expressible in large quantities from *E. coli*, with either the human sequence[44], its murine orthologue[30,32] or the single-domain orthologue *M. gryphiswaldense* CRBN isoform 4[45,46], which all crystallize routinely to sub-2 Å resolution. However, the TBD domain alone lacks a significant portion of the full-length CRBN protein and therefore is not representative for studies of ternary complexes which often involve interactions with the missing Lon domain.

To address these limitations, we set out to develop a truncated CRBN construct containing both TBD and Lon domains, that would conveniently express soluble and stable protein from *E. coli* with comparable functionality to the full-length wild-type CRBN. To this end, we rationally designed fifteen new constructs using three main strategies: (i) partial deletions of the HB domain (constructs **1**–**4**), (ii) introducing stabilizing mutations in a construct lacking the DDB1-interacting region (constructs **5**–**10**), and (iii) deletion of the full HB domain (constructs **11**–**15**) (Fig. 1a, Table 1). In detail, first we removed the DDB1-binding region of the HB domain, containing residues 188-248, which in absence of DDB1 exposes a hydrophobic patch prone to inducing aggregation of CRBN, resulting in constructs **1** and **2** with GSG or GGGGS linkers, respectively. Further truncations of the remaining HB domain were guided by secondary structure boundaries. Construct **3** contains two C-terminal alpha-helices from the HB domain spanning residues 277–317 with a GGGGS linker and construct **4** spans HB residues 302–317, where Ile305-Lys317 was mutated to a polyAla helix. The removal of portions of the HB domain resulted in previously buried residues becoming solvent-exposed, in some cases compromising the solubility of the construct. In our second approach, we used the Protein Repair One-Stop Shop (PROSS) server[47] to identify amino acid substitutions that would positively contribute to protein stability and solubility. Constructs **5**–**10** were designed by manually curating the results of the PROSS analysis to exclude any residues within, and in the vicinity of, the IMiD-binding site that would engage in protein-protein interactions during ternary complex formation[18–26]. Thirdly, the HB domain was completely removed in constructs **11**–**15** to contain only the TBD and Lon domains connected by an assortment of flexible linkers and the C366S mutation previously introduced to improve crystallization of the TBD domain[44]. In all constructs, glycine and serine-rich linkers were chosen for their flexibility[48] with the length requirement predicted for each truncation by analysis of published CRBN crystal structures.

To assess protein expression and solubility, we cloned all constructs with N-terminal His[6] or His[6] coupled to maltose-binding protein (MBP) affinity tags and performed Ni-NTA affinity pull-down assay from lysates of cells transformed with each construct (Fig. 1b). All His[6]-MBP-tagged constructs showed detectable immobilized protein levels whereas only constructs **2** and **5**–**8** immobilized above the detection limit in the absence of the MBP tag. Importantly, the stabilizing mutations introduced based on PROSS analysis proved to have beneficial effects, with the best-expressing construct (construct **8**, hereafter referred to as CRBN[midi]) incorporating 12 mutations. CRBN[midi] protein was subsequently expressed and purified in a large scale to assess its yield and homogeneity. The purification involved Ni-NTA affinity chromatography, desalting, tag cleavage, and reverse Ni-NTA affinity chromatography followed by size-exclusion chromatography (SEC) which yielded a pure monodisperse sample as confirmed by SDS-PAGE and SEC (Supplementary Fig. 1) resulting in 1.5 mg of purified protein per 1 L of culture. Next, we successfully crystallized CRBN[midi] protein and determined its crystal structure to a resolution of 3.11 Å (Fig. 1c, Supplementary Table 1). The fold of CRBN[midi] superposes well with the corresponding portion of CRBN[ΔN]:DDB1:lenalidomide (PDB ID: 4TZ4) with a RMSD of 1.27 Å (over 232 out of 275 Cα atoms) indicating the new construct adopts a biologically relevant fold equivalent to that of the wild-type protein. As with CRBN[ΔN]:DDB1, the structure shows CRBN[midi] co-purified with a $Zn^{2+}$ ion bound to the zinc finger of the TBD. Interestingly, the protein in the structure is observed to adopt the

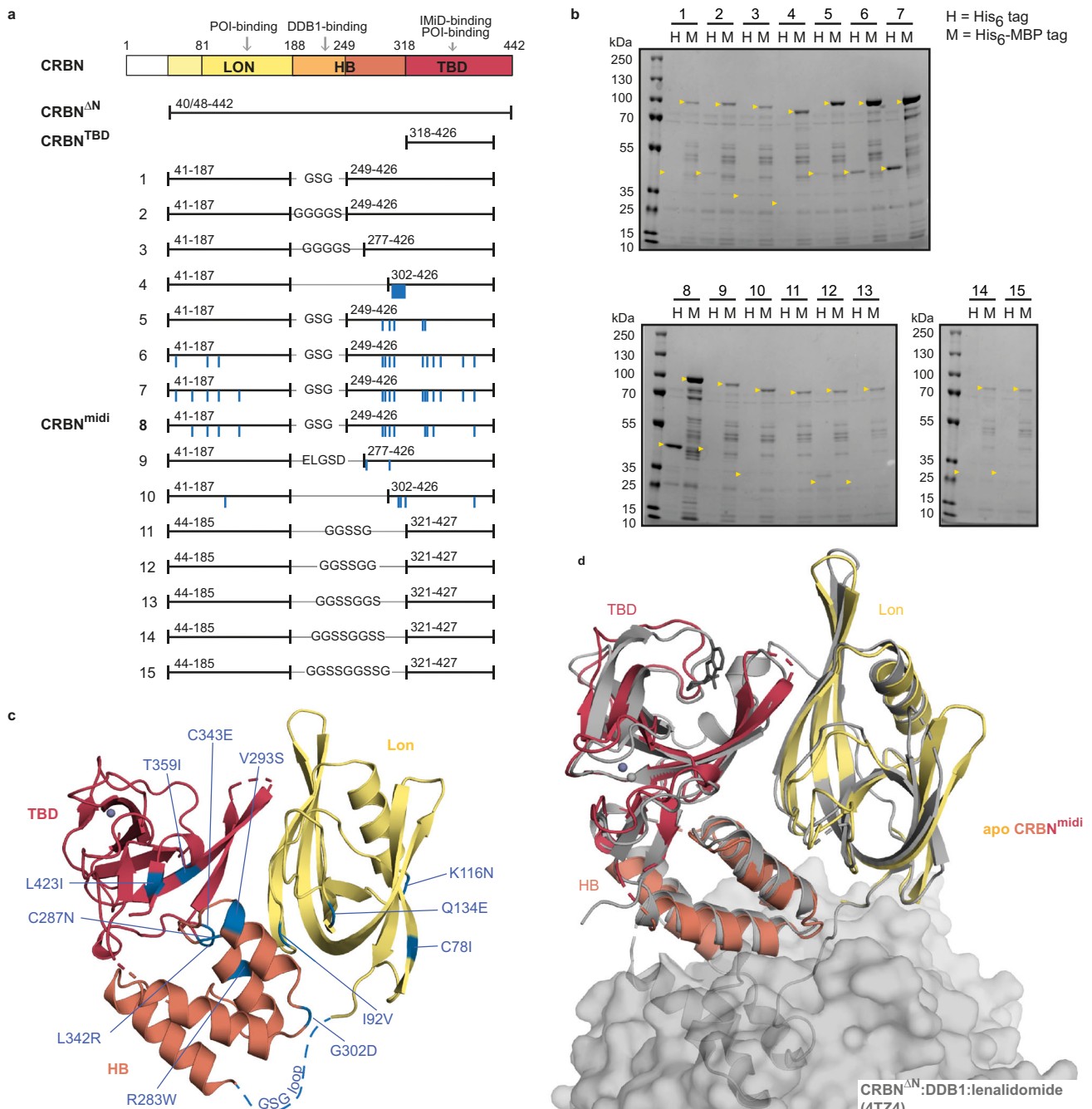

**Fig. 1 | Design and characterization of CRBN^midi. a** Domain composition of full-length CRBN (top), constructs previously used for structural and biophysical characterization, CRBN^ΔN and CRBN^TBD, and constructs **1**–**15** designed in this study. The most promising construct **8** was designated as CRBN^midi. Mutated residues are indicated as blue lines. **b** Ni-NTA affinity pull-down assay of lysates from *E. coli* BL21(DE3) cells expressing candidate constructs **1**–**15** with an N-terminal His$_6$ (H) or His$_6$-MBP (M) tag. The expected size of each construct is marked with a yellow arrow. The experiment was performed as one replicate. Uncropped gels are provided as a Source Data file. **c** Crystal structure of CRBN^midi in the apo state, containing Lon (yellow), HB (orange) and TBD (red) domains. The mutated residues are indicated (blue), the unresolved region containing the GSG linker is shown as blue dashed line, and Zn$^{2+}$ is shown as a purple sphere. **d** Superposition of apo CRBN^midi (yellow, orange, red) with CRBN^ΔN:DDB1 bound to lenalidomide (grey, PDB ID: 4TZ4[10]). DDB1 is shown as a grey surface.

closed conformation, despite the absence of a small-molecule ligand. The compact closed conformation was likely selected for by the crystallization process as, unlike for ligand-bound CRBN^midi which produces large numbers of hits, we only obtained a single crystal for apo CRBN^midi indicating that apo CRBN^midi is recalcitrant to crystallization in its flexible open conformation. In addition, as CRBN^midi is purified in the absence of DDB1, the additional stabilisation of the open confirmation through interactions between the sensor loop (residues 341–361) and DDB1(residues -776–780) that was previously observed

by Watson et al.[25] cannot take place in this structure, and is likely a contributing factor to observing the closed state.

## CRBN^midi binary studies reveal molecular insights into ligand recognition

We next assessed the suitability of CRBN^midi for accessing protein-ligand binary complexes. We determined the co-crystal structures of CRBN^midi in complex with mezigdomide or lenalidomide to a resolution of 2.19 and 2.50 Å, respectively (Supplementary Table 2).

**Table 1 | Design of CRBN constructs**

| Construct ID | N-terminal residue range | Linker | C-terminal residue range | Mutations |
|---|---|---|---|---|
| 1 | 41–187 | GSG | 249–426 | – |
| 2 | 41–187 | GGGGS | 249v426 | – |
| 3 | 41–187 | GGGGS | 277–426 | – |
| 4 | 41-187 | – | 302–426 | I305-K317 -> polyA |
| 5 | 41–187 | GSG | 249–426 | R283W, V293S, G302D, L340M, C343E |
| 6 | 41–187 | GSG | 249–426 | T58S, I92V, K116N, R283W, C287N, V293S, G302D, L340M, C343E, T359I, C366K, S410R, L423I |
| 7 | 41–187 | GSG | 249–426 | T58S, C78I, I92V, K116N, Q134E, R283W, C287N, V293S, G302D, L340M, L342R, C343E, T359I, C366K, S410R, L423I |
| 8 | 41–187 | GSG | 249–426 | C78I, I92V, K116N, Q134E, R283W, C287N, V293S, G302D, L342R, C343E, T359I, L423I |
| 9 | 41–187 | ELGSD | 277–426 | P277T, V293H |
| 10 | 41–187 | – | 302–426 | V128R, A304K, Q306D, C310H, L423Q |
| 11 | 44–185 | GGSSG | 321–427 | C366S |
| 12 | 44–185 | GGSSGG | 321–427 | C366S |
| 13 | 44–185 | GGSSGGS | 321–427 | C366S |
| 14 | 44–185 | GGSSGGSS | 321–427 | C366S |
| 15 | 44–185 | GGSSGGSSG | 321–427 | C366S |

The CRBN[midi]:mezigdomide structure revealed a binding mode consistent with a previously published CRBN:DDB1:mezigdomide cryo-EM structure (PDB ID: 8D7U) with superposition yielding RMSD of 0.67 Å (over 259 out of 306 Cα atoms) (Fig. 2a). In addition, the crystal structure of mezigdomide-bound CRBN[midi] revealed a non-covalent interaction between the fluorine atom of the ligand and Asp149 of the Lon domain which was previously undetected in the CRBN:DDB1:mezigdomide cryo-EM structure[25] likely due to lower map quality around the ligand binding site (Fig. 2a middle panel). As mezigdomide interacts with both the TBD and Lon domains of CRBN, the CRBN[midi]:mezigdomide structure illustrates the importance of studying binary interactions using a construct containing both TBD and Lon domains in contrast to the CRBN[TBD] construct often used for binary studies. Similarly to the CRBN[midi]:mezigdomide structure, the crystal structure of CRBN[midi] bound to lenalidomide shows the closed conformation (Fig. 2b) and superposition with CRBN[ΔN]:DDB1:lenalidomide (PDB ID: 4TZ4) indicates conserved ligand binding (RMSD of 1.22 Å over 227 out of 288 Cα atoms).

Furthermore, to assess the suitability of CRBN[midi] to study alternative, non-thalidomide scaffolds, we solved the crystal structure of CRBN[midi] in complex with compound **1**, a phenyl-glutarimide based binder[16], to a resolution of 1.95 Å (Fig. 2c, Supplementary Table 2). Superposition with CRBN[ΔN]:DDB1:lenalidomide (PDB ID: 4TZ4) yielded an RMSD of 1.37 Å over 262 out of 317 Cα atoms. While compound **1** used for crystallization was a racemic mixture, only the (R) enantiomer fit the electron density well. The two protomers in the asymmetric unit capture two different conformations of the N-methylacetamide group in compound **1** indicating flexibility (Fig. 2c, 2nd panel from top). A lack of conformational restriction in this region may be desirable for this compound, as the N-methylacetamide represents a key exit vector location in compound **1** for building out into a bifunctional molecule such as a PROTAC[16]. Comparisons of the crystal structures of CRBN[midi] bound to compound **1** and lenalidomide reveal that the glutarimide in compound **1** adopts a near identical binding mode to the equivalent moiety in lenalidomide (Fig. 2c, 3rd panel from top). Together, these high-resolution co-crystal structures validate CRBN[midi] as an enabling construct for X-ray crystallographic structure-based binder design.

CRBN transitions from an open to a closed conformation upon ligand binding, with major hinge bending between the TBD and Lon domains[19,21,25]. Our extensive engineering of the linker region between these two domains in CRBN[midi] prompted us to next conduct biophysical experiments to investigate the behaviour of CRBN[midi] in solution. Whilst both apo and ligand-bound CRBN[midi] crystallized in the closed conformation, we used small-angle X-ray scattering (SAXS) coupled to in-line SEC to analyse the domain rearrangement of CRBN[midi] in solution (Fig. 2e, Supplementary Fig. 2, Supplementary Table 3). In the absence of a binder, the radius of gyration ($R_G$) for CRBN[midi] was 26.73 Å, corresponding well to the predicted $R_G$ of 25.65 Å for apo CRBN[midi] in its open conformation, as modelled based on the full-length CRBN cryo-EM structure (PDB ID: 8CVP)[25]. A theoretical scattering curve for CRBN[midi] modelled in the open conformation is also consistent with the experimental SAXS scattering curve of apo CRBN[midi] (Supplementary Fig. 2). In the presence of 400 μM ligand, we observed a $R_G$ of 22.67 and 22.49 Å for mezigdomide- and lenalidomide-bound CRBN[midi], respectively, which were significantly smaller than in the absence of ligand (26.73 Å), and consistent with the calculated $R_G$ of closed CRBN[midi] (23.76 Å). Furthermore, experimental scattering curves for CRBN[midi] in the presence of binders were consistent with theoretical scattering curves calculated for closed CRBN[midi] (Supplementary Fig. 2).

Due to the significant change to the $R_G$ of CRBN[midi] observed on binding to lenalidomide and mezigdomide, we theorised that this SAXS approach could be used to validate small molecule binding to CRBN[midi]. To investigate this further we selected two Boc-protected cyclic-IMiD di-peptides as example epitopes of the natural degron of CRBN; Boc-VcN and Boc-AcQ (Fig. 2d) for CRBN[midi] SAXS analysis. We collected CRBN[midi] SAXS curves in the presence of 200 μM Boc-VcN and Boc-AcQ and observed contracted $R_G$ values of 23.41 and 24.29 Å, respectively, relative to apo CRBN[midi] (26.73 Å). This is consistent with stabilisation of the closed, ligand-bound conformation of CRBN[midi].

To compare the extent of flexibility of the apo and ligand-bound protein we used the dimensionless Kratky plot[49], a representation of the SAXS data normalised for scattering intensity and $R_G$ (Fig. 2e). It indicates that apo CRBN[midi] shows significant flexibility and can access a larger conformational landscape, evidenced by the tail of the bell curve trailing higher than that of the ligand-bound CRBN[midi] curves. In contrast, the curves for the ligand-bound return to the baseline and show a peak maximum close to the Guinier-Kratky point, indicating a rigid, globular conformation. This is consistent with the findings from

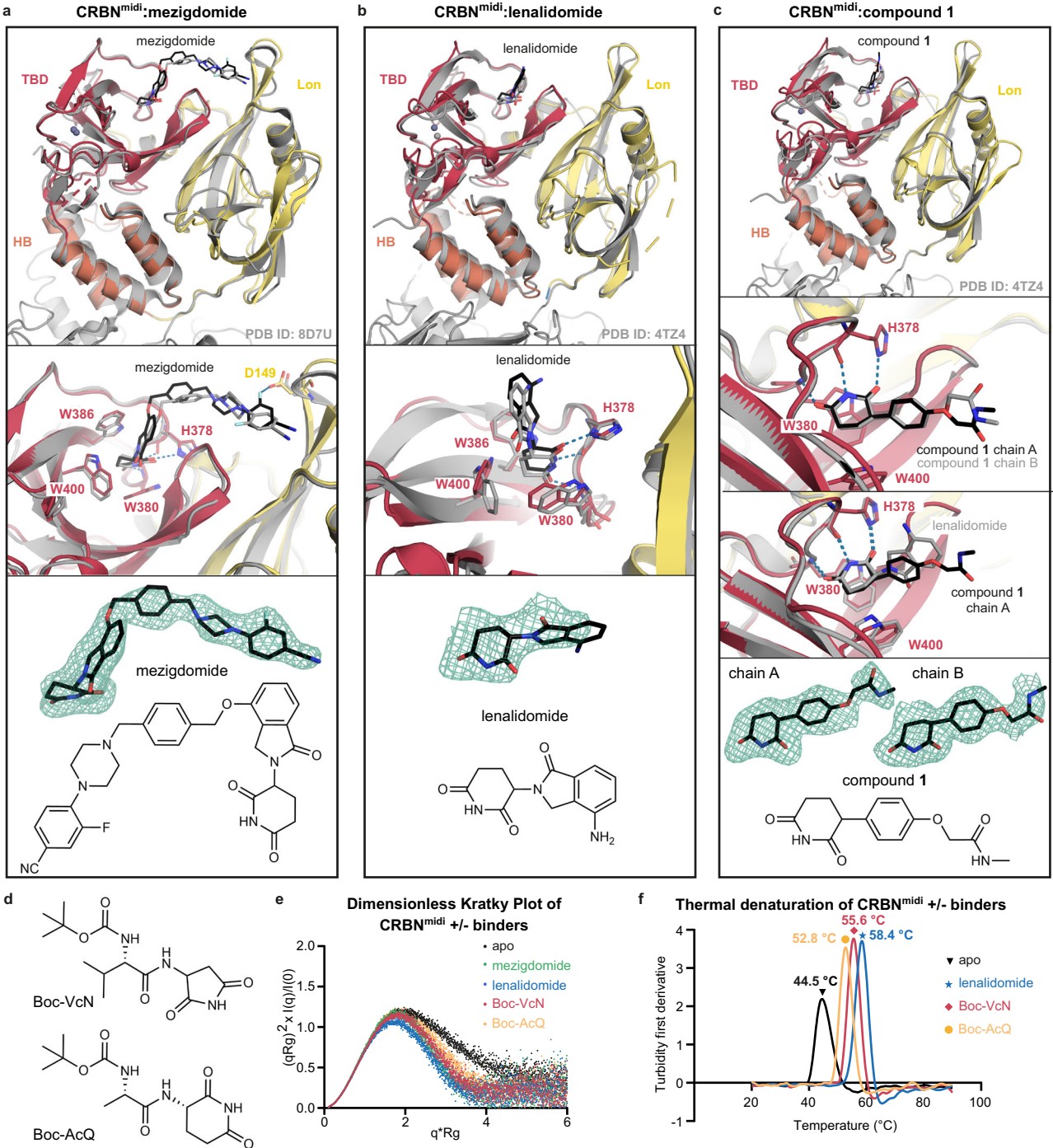

**Fig. 2 | Structural and biophysical analysis of binary protein-ligand complexes using CRBN^{midi}. a** Crystal structure of CRBN^{midi} (domains coloured as in Fig. 1) bound to mezigdomide (shown as black sticks) superposed with a cryo-EM structure of CRBN:DDB1:mezigdomide (grey, PDB ID: 8D7U). **b** Crystal structure of CRBN^{midi} (domains coloured as in Fig. 1) bound to lenalidomide (shown as black sticks) superposed with a crystal structure of CRBN^{ΔN}:DDB1:lenalidomide (grey, PDB ID: 4TZ4). **c** Crystal structure of CRBN^{midi} (domains coloured as in Fig. 1) bound to compound **1** (shown as black sticks) superposed with a crystal structure of CRBN^{ΔN}:DDB1:lenalidomide (grey, PDB ID: 4TZ4, top panel). Overlay of compound **1** in the two protomers in the asymmetric unit (chain A and chain B) shown in 2^{nd} panel from the top, overlay of chain A and CRBN^{midi}:lenalidomide shown in 3^{rd} panel from the top. **a-c** Overall fold shown as cartoon (top); detail of the ligand binding site (middle) with residues involved in ligand binding shown as sticks and potential H-bonds shown as blue dashed lines; polder OMIT (Fo-Fc) map of the ligands contoured to 3 σ shown as green mesh, alongside the chemical structures of the compounds (bottom panel). **d** Chemical structures of Boc-VcN and Boc-AcQ. **e** Dimensionless Kratky Plot generated from SAXS data of apo CRBN^{midi} (black) and CRBN^{midi} bound to mezigdomide (green), lenalidomide (blue), Boc-VcN (red), or Boc-AcQ (orange). **f** First derivative of turbidity of thermal denaturation for CRBN^{midi} in the absence (black) or presence of binders lenalidomide (blue), Boc-VcN (red), or Boc-AcQ (orange).

Watson et al. that show that apo cereblon can readily access open conformations in cryo-EM studies[25]. Furthermore, molecular dynamics (MD) simulation on our CRBN^{midi}:mezigdomide crystal structure (PDB ID: 8RQ8, this publication) alongside simulation of the CRBN^{ΔN}:mezigdomide cryo-EM structure (PDB ID: 8D7U)[25] was next performed to computationally investigate and compare the dynamics of the protein residues and the ligand interaction profile in both protein constructs. The Root Mean Square Fluctuations (RMSF) of the

residues and protein-ligand interaction profiles as observed from the MD simulations indicated that CRBN$^{midi}$ behaves similarly to the CRBN$^{\Delta N}$ construct (Supplementary Fig. 3, see Supplementary note for details). These data evidence overall compaction of CRBN$^{midi}$ protein undergoing the open-closed conformational rearrangement upon ligand binding, in a manner consistent with wild-type full-length CRBN, and exemplify the sensitivity of SAXS for monitoring binding of CRBN ligands which can modulate this open-closed equilibrium.

To gain further insights into the biophysical thermodynamics of the protein-ligand binding equilibria, we performed differential scanning fluorimetry (DSF) and isothermal titration calorimetry (ITC) studies with CRBN$^{midi}$. Using DSF, we compared the thermal denaturation curves of ligand-bound and unbound CRBN$^{midi}$ with those of CRBN$^{\Delta N}$:DDB1$^{\Delta BPB}$ (Fig. 2f, Supplementary Fig. 4a). CRBN$^{midi}$ was significantly thermally stabilised in the presence of binder, with an apo melting temperature ($T_m$) of 44.5 °C that increased by 8-14 °C in the presence of 100 μM binder (lenalidomide 58.4 °C, Boc-VcN 55.6 °C, Boc-AcQ 52.8 °C) (Fig. 2f). In comparison, CRBN$^{\Delta N}$:DDB1$^{\Delta BPB}$ showed a $T_m$ of 53 °C in the absence of ligand and $T_m$ of 55 °C in the presence of lenalidomide ($\Delta T_m = 2$ °C) (Supplementary Fig. 4a). This large $\Delta T_m$ upon ligand binding qualifies CRBN$^{midi}$ as a suitable construct for high-throughput screening of potential CRBN binders and modulators by DSF.

Next, we obtained binary binding affinity measurements by ITC. Titration of lenalidomide into CRBN$^{midi}$ resulted in a conventional sigmoidal curve with an exothermic profile, and data fitting yielded a $K_D$ of 2.9 μM and ΔH of -24 kJ/mol (Supplementary Fig. 4b). For reference, ITC titration data have previously been reported for lenalidomide against CRBN$^{TBD}$ ($K_D$ of 19 μM, ΔH of -21.8 kJ/mol) and CRBN:DDB1 ($K_D$ of 0.6 μM, ΔH of 10.5 kJ/mol)[50]. The differences in binding affinity and enthalpy observed are consistent with the use of different protein constructs and with differences in experimental conditions, meaning absolute $K_D$ and ΔH values are not directly comparable. Nonetheless, our data benchmarks CRBN$^{midi}$ against widely used current constructs and qualifies CRBN$^{midi}$ as relevant and suitable for ITC analysis of protein-ligand binding. Overall, we show that CRBN$^{midi}$ enables in-depth structural and biophysical analysis of binary interactions via a wide range of biophysical and structural techniques thus qualifying it as suitable for library screening towards the identification and characterization of new binders.

## High-resolution crystal structures and biophysical characterization of degrader ternary complexes enabled by CRBN$^{midi}$

We set out to exemplify the utility of CRBN$^{midi}$ for enabling co-crystal structures of ternary complexes formed by molecular glue or PROTAC degraders. We first determined the crystal structure of CRBN$^{midi}$ in complex with mezigdomide and the second zinc-finger of neo-substrate Ikaros (IKZF1$^{ZF2}$) to a resolution of 2.15 Å (Fig. 3a–c, Supplementary Table 1). Superposition of our X-ray structure with the cryo-EM structure of CRBN:DDB1:mezigdomide:IKZF1$^{ZF1-2-3}$ (PDB ID: 8D7Z)[25] shows a highly consistent binding mode of both IKZF1 and mezigdomide engaged in the ternary complex with CRBN (RMSD of 0.85 Å over 245 out of 293 Cα atoms) (Fig. 3b). Importantly, the higher resolution of our crystal structure and greater quality of the density map at the binding interface allowed us to unambiguously observe several protein-ligand and protein-protein interactions, including contacts that had not previously been identified (Fig. 3c). An extended network of mezigdomide-CRBN contacts is observed. At the phthalimide-end, the ligand's glutarimide moiety interacts via three canonical hydrogen bonds with His378$^{CRBN}$ and Trp380$^{CRBN}$, while the oxoisoindoline carbonyl group forms a hydrogen bond with the side chain NH$_2$ of Asn351$^{CRBN}$. Mezigdomide adopts a characteristic U-like shape around its central phenyl ring, wrapped around Pro352 and His353 at the tip of the beta-hairpin sensor loop (residues 341–361), conformationally stabilized through several π-π stacks and hydrophobic interactions

(Fig. 3c). At the other end, the piperidine-benzonitrile moiety of mezigdomide fits into a hydrophobic pocket defined on one side by Phe102, Phe150 and Ile152 from the CRBN's Lon domain, and on the other side by the tip of the sensor loop (Fig. 3c). In contrast to the extensive contacts between mezigdomide and CRBN, no further interactions are observed between mezigdomide and IKZF1$^{ZF2}$ beyond the known stacking of the neo-substrate Gly-loop on top of the iso-indolinone ring, consistent with the molecular glue degrader binding exclusively to CRBN and not IKZF1 at the binary level[19,21]. In addition, we observed several direct protein-protein contacts mediated by the degrader, also not all previously observed: Asn148$^{IKZF1}$, via a hydrogen bond from its backbone carbonyl to the side chain NH$_2$ of Asn351$^{CRBN}$, and through side chain packing with His353$^{CRBN}$; Gln149$^{IKZF1}$, via a hydrogen bond from its backbone carbonyl to the side chain NH$_2$ of Asn351$^{CRBN}$, and side chain contacts with Tyr355$^{CRBN}$; and Cys150$^{IKZF1}$, via a hydrogen bond from the backbone carbonyl to the side chain NH of Trp400$^{CRBN}$ (Fig. 3c). In our structure, Cys150$^{IKZF1}$ is not within hydrogen bonding distance of His397$^{CRBN}$, unlike the cryo-EM structure[25]. Moreover, compared to our binary CRBN$^{midi}$:mezigdomide structure (Fig. 2a), the fluorine of mezigdomide does not engage Asp149 in the ternary structure with IKZF1$^{ZF2}$. Overall, our co-crystal structure has allowed us to detail many protein-ligand and protein-protein interactions within the ternary complex at a resolution beyond that which has been previously achieved.

We next solved the ternary crystal structure of CRBN$^{midi}$ in complex with PROTAC CFT-1297 and BRD4$^{BD2}$ to a resolution of 2.9 Å (Fig. 3d–g, Supplementary Table 1). We chose compound CFT-1297 as a BET PROTAC degrader for crystallographic studies because it is reported to exhibit positive cooperativity of ternary complex formation, and was previously characterized by hydrogen-deuterium exchange mass spectrometry (HDX-MS)[27], providing some low-resolution structural information. The crystal structure contains two protomers in the asymmetric unit, indicating flexibility in the ternary complex. The two protomers show a slightly different bound pose of the BRD4$^{BD2}$ molecules relative to CRBN, caused by a tilt of 27° of the PROTAC around the piperidine moiety that connects the CRBN binder and the alkyl linker (Fig. 3e). The bromodomain docks sideways on top of the CRBN TBD, an orientation imposed by the long rigid alkyl linker of the PROTAC. The surface of the bromodomain most buried within the complex comprise the region from the end of the Z-helix to the middle of the ZA-loop, including the αZ' helix from the ZA loop (residues 362–378). This region packs against the central linker region of the PROTAC at one end, and the CRBN TBD between the sensor loop and residues 390–396 at the other end, where a hydrogen bond is predicted between the side chains of Tyr372$^{BRD4}$ and Gln390$^{CRBN}$ (Fig. 3f). The remainder of the ZA loop (residues 379–400) and the whole BC-loop are solvent-exposed within the ternary complex structure, as well as the Lon domain of CRBN which is also fully solvent-exposed (Fig. 3d). These observations are inconsistent with the HDX-MS results from Eron et al. where shielding of the BC loop and Lon domain in Brd4$^{BD1}$ and CRBN, respectively, were observed in the ternary complex[27], suggesting a differential mode of recognition for BD1 vs. BD2. Additional protein-protein hydrophobic contacts are also observed as the C-helix of the bromodomain (residues 438–449) is found packed against CRBN at the opposite side of the TBD, with the side chains of Glu438$^{BRD4}$ and Arg373$^{CRBN}$ at this interface located within a potential salt-bridge distance. Overall, our co-crystal structure reveals at high resolution an extensive network of interactions mediating the recognition mode of a CRBN:PROTAC:Brd4 ternary complex, and suggests a distinct ternary complex recruitment mode of the two bromodomains of Brd4 to CRBN by CFT-1297. Furthermore, by capturing two orientations of BRD4$^{BD2}$ this crystal structure provides structural insight into complex flexibility that could be challenging to visualize by other techniques, such as cryoEM, in which flexibility can limit resolution.

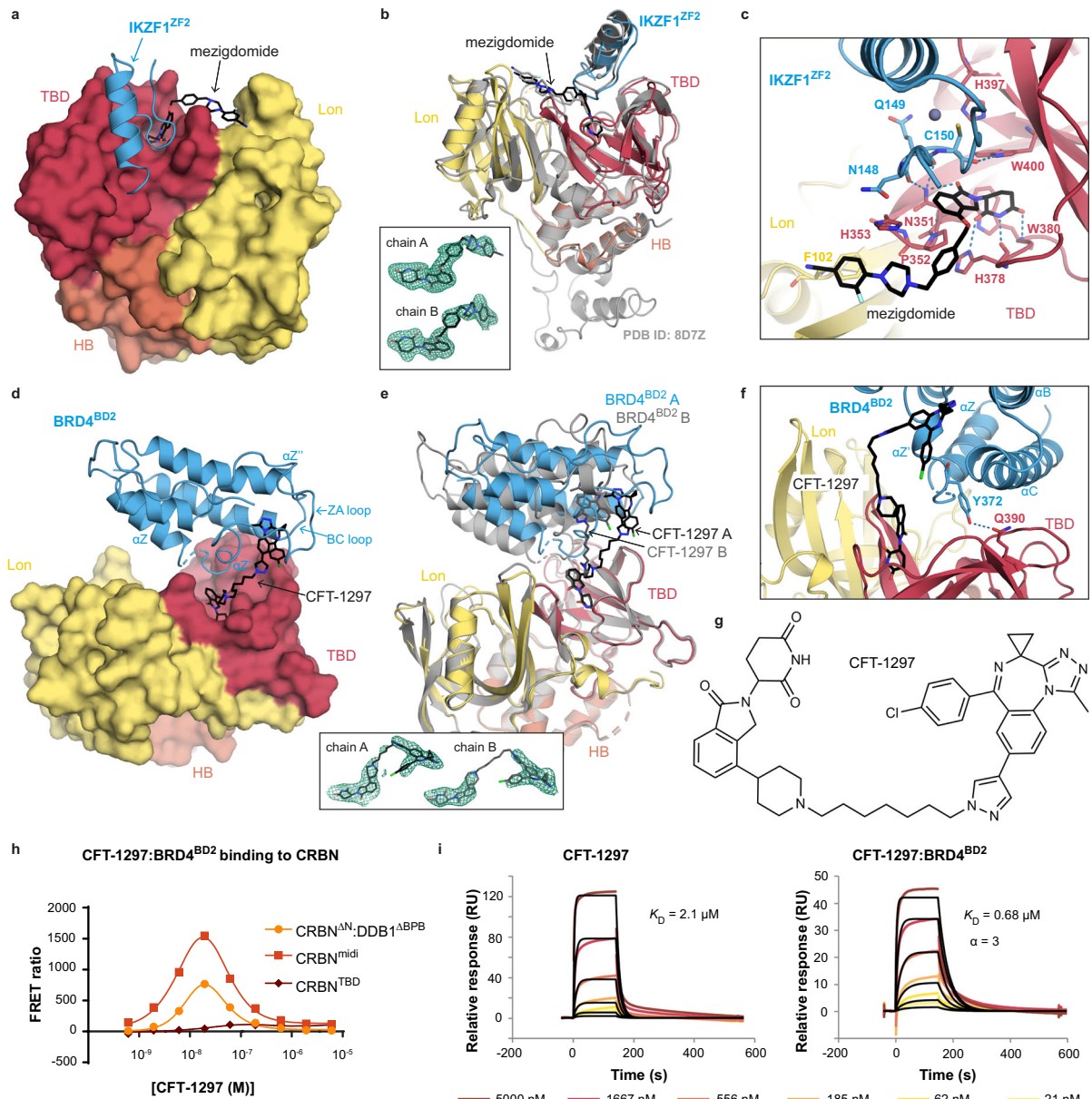

**Fig. 3 | Ternary complex characterization using CRBN^midi. a–c** Crystal structure of CRBN^midi in complex with mezigdomide and IKZF1^ZF2. **a** CRBN^midi is shown in surface representation, mezigdomide is shown as black sticks and IKZF1^ZF2 is shown as blue cartoon. **b** Superposition of the CRBN^midi:mezigdomide:IKZF1^ZF2 crystal structure (coloured as in A) with the cryo-EM structure of CRBN:DDB1: mezigdomide:IKZF1^ZF1-2-3 (PDB ID: 8D7Z, grey, DDB1 not shown). Polder OMIT (Fo-Fc) map of mezigdomide in protomers A and B contoured to 3 σ is shown in the inset. **c** Close-up of the ligand-protein and protein-protein interaction network in **b**. Hydrogen bonding interactions in the CRBN^midi:mezigdomide:IKZF1^ZF2 interface are shown as blue dashed lines. **d-g** Crystal structure of CRBN^midi in complex with CFT-1297 and BRD4^BD2. **d** CRBN^midi is shown in surface representation, CFT-1297 is shown as black sticks and BRD4^BD2 is shown as blue cartoon. **e** The two protomers in the asymmetric unit, chain A coloured, chain B in grey. Polder (OMIT) (Fo-Fc) map of CFT-1297 in chains A and B contoured to 3 σ shown in the inset. **f** Close-up of the interface between BRD4^BD2 (chain B) and CRBN^midi, residues Tyr372^BRD4 and Gln390^CRBN are shown as sticks with a likely hydrogen bond shown as a blue dashed line. **g** Chemical structure of CFT-1297. **h** TR-FRET traces monitoring ternary complex formation of BRD4^BD2 and different constructs of CRBN in the presence of a dilution series of CFT-1297. **i** CRBN^midi-on-chip SPR sensograms measuring binding of CFT-1297 alone (left) and CFT-1297 pre-incubated with BRD4^BD2 (right).

Across the apo, binary, and ternary crystal structures presented in this study, CRBN^midi has been shown to crystallize in a range of different space groups with crystal contacts mediated by different regions whilst maintaining a consistent fold (Supplementary Tables 1, 2; Figs. 2, 3). Furthermore, crystal contacts do not appear to significantly influence ligand binding conformation or ternary complex orientation in any of the structures solved in this study (Supplementary Fig. 5).

To establish the utility of CRBN^midi to biophysical studies of PROTAC ternary complexes in solution, we next performed TR-FRET

and SPR binding assays. First, we monitored the proximity between BRD4^BD2 and CRBN^midi upon increasing concentrations of CFT-1297 using TR-FRET, alongside CRBN^TBD and CRBN^ΔN:DDB1^ΔBPB to allow benchmarking of our construct (Fig. 3h). CRBN^ΔN:DDB1^ΔBPB and CRBN^midi produced the expected bell-shaped profile exhibiting the characteristic hook effect at high concentration of bifunctional molecule, and with near identical maximal concentrations (17 nM *vs.* 21 nM, respectively) and comparable broadness of the curves. In contrast, no significant ternary complex formation was detected for the CRBN^TBD

construct, highlighting the importance of using a CRBN construct that includes the Lon domain for biophysical characterization of degrader ternary complexes. With formation of the ternary complex CRBN$^{midi}$:CFT-1297:BRD4$^{BD2}$ confirmed by TR-FRET, we next aimed to quantitatively measure the binding affinity, cooperativity and kinetic stability of this species using an SPR ternary complex assay conceptually similar to the one previously developed by us with VHL[37] and by Bonazzi et al. and Ma et al. for CRBN:DDB1[26,29]. First, we immobilised CRBN$^{midi}$ on a Ni-NTA SPR chip via the His-tag and monitored the binding of CFT-1297 to CRBN$^{midi}$ in the absence or presence of saturating amounts of BRD4$^{BD2}$ (Fig. 3i, and Supplementary Table 4). We observed a 3-fold enhanced affinity of CRBN for the PROTAC:BRD4$^{BD2}$ binary complex when compared to that for the PROTAC alone (ternary $K_D = 0.68\,\mu M$ *vs.* binary $K_D = 2.1\,\mu M$) and a corresponding increase in the stability of ternary *vs.* binary complex as measured from dissociation kinetics ($t_{1/2} = 31\,s$ *vs.* 9 s). This positive cooperativity ($\alpha = 3$) and increased durability of CRBN E3 ligase engagement suggest favourability of the extensive network of intra-complex contacts observed in our ternary co-crystal structure.

In summary, targeted protein degradation has shown considerable promise as a route for therapeutic intervention. Co-opting of the E3 ligase CRBN is an established viable approach by using molecular glue or PROTAC degraders to productively engage with target proteins to trigger their rapid ubiquitination and degradation in vivo. The formation of the ternary complex is the key mechanistic step in the mechanism of action of small molecule degraders. It is established that characteristics such as affinity and half-life of the ternary complex correlate well with degradation fitness[28,30,37,38], and potentiating ternary complex affinity, stability and favourability to target ubiquitination provides a powerful optimization strategy[31–34,42]. This goal can be expedited through biophysical and structural insights of CRBN recruitment, ushering structure-guided design of additional ligands and improved protein degraders. Our CRBN$^{midi}$ construct provides a reagent that extends and enables future endeavours in several ways. First, the nature of our engineering, including both TBD and Lon domains in a single construct, addresses and solves many limitations of current constructs, as illustrated by convenient high-level expression in *E. coli*, solubility, and stability as monomeric protein without the need to co-express with DDB1, as well as apparent readiness to crystallize in ligand-bound forms. Second, we exemplify the utility of CRBN$^{midi}$ in allowing X-ray crystallography of CRBN ternary complexes that had not been crystallized before and at higher resolution than previously attained with other constructs through either X-ray or cryo-EM investigations[20,25,27]. Third, we demonstrate the suitability of CRBN$^{midi}$ for biophysical ligand binding studies in solution, and provide a robust benchmark against wild-type functionality through a variety of assay read-outs and settings via techniques including SAXS, DSF, TR-FRET, ITC and SPR. The soluble, stable, and readily producible CRBN$^{midi}$ construct that we here disclose and validate could be useful not only for enabling structure-based screens and designs for therapeutic applications but also as a facile molecular tool for probing native interactions and structure-function properties of CRBN.

Moving forward, there are many avenues to pursue leveraging our construct and the biophysical and structural assays described here and beyond to accelerate ligand screening and design, and to illuminate molecular mechanisms. The robustness and convenience of CRBN$^{midi}$ and its propensity to yield high-resolution liganded co-crystal structures as evidenced in this study open the door to a wide range of refined strategies for the design and optimization of protein degradation therapies. As with any tool, the utility and applicability of CRBN$^{midi}$ must be duly considered for each application, and there remain caveats. For example, to aid crystallizability, we have deleted the flexible 40 N-terminal residues that, albeit not essential, have been shown to contribute to stabilizing the closed conformation of CRBN[25]. The absence of DDB1 from complexes of CRBN$^{midi}$ could also limit the utility of this construct in the search for allosteric ligands of CRBN, particularly where binders stabilizing either the open or closed confirmation of CRBN or binders at the CRBN-DDB1 interface are sought. Similarly, whilst CRBN$^{midi}$ can help to determine immediate interactions within the ternary complexes, it will remain a surrogate for studying the native CRL4$^{CRBN}$ E3 ligase complexes. Nonetheless, we anticipate that the data reported in this study and the open-access availability of the construct e.g. via Addgene, will aid broad uptake by the community and lead to many advances in the targeted protein degradation field, ultimately accelerating the discovery of degrader therapeutics.

## Methods

### Chemical reagents
All chemical reagents were procured from external suppliers. Mezigdomide (CC-92480) was purchased as solid from SelleckChem catalogue number S8975. Lenalidomide was purchased from Combi-Blocks catalogue number OR-2352. CFT-1297 was synthesized to order by Aragen Life Sciences. 2-(4-(2,6-dioxopiperidin-3-yl)phenoxy)-*N*-methylacetamide (compound **1**) was synthesised as previously reported by Min et al.[16]. *Tert*-butyl ((2S)-1-((2,5-dioxopyrrolidin-3-yl)amino)-3-methyl-1-oxobutan-2-yl)carbamate (Boc-VcN) was synthesised as per Yamaguchi et al.[51], and *tert*-butyl ((S)-1-(((S)-2,6-dioxopiperidin-3-yl)amino)-1-oxopropan-2-yl)carbamate (Boc-AcQ) was synthesized as per Ichikawa et al.[28].

### Cloning
The stabilizing mutations in human CRBN (UniProt: Q96SW2) were designed using the PROSS server accessed in February 2022[47]. DNA encoding the CRBN constructs was synthesized as gBlock Gene Fragments by Integrated DNA Technologies and inserted into a pNIC28-Bsa4 (Addgene #26103) vector using ligation-independent cloning, as well as into a pNIC28-Bsa4 with an additional maltose-binding protein (MBP) tag inserted between the His$_6$ and TEV (Tobacco Etch Virus) protease cleavage site. The final CRBN$^{midi}$ construct contained amino acids 41-187 followed by a Gly-Ser-Gly linker and amino acids 249-426 with single-point mutations C78I, I92V, K116Q, Q134E, R283W, C287N, V293S, G302D, L342R, C343E, T359I and L423I.

### Small-scale test protein expression
The His$_6$- and His$_6$-MBP-tagged CRBN fusion proteins were expressed in BL21(DE3) *E. coli* cells grown in 3 mL of Lysogeny broth (LB) supplemented with 50 µg/mL kanamycin. After 4.5 h at 37 °C, protein expression was induced with 0.5 mM isopropyl β-d-1-thiogalactopyranoside (IPTG) at 18 °C overnight. After harvesting, the cells were lysed in buffer containing 20 mM HEPES pH 8.0, 500 mM NaCl, 10 mM imidazole, 1 mM tris(2-carboxyethyl)phosphine hydrochloride (TCEP), 5 mM MgCl$_2$, cOmplete Protease Inhibitor (Roche), 0.5 mg/mL lysozyme, and 10 µg/mL DNase I (Roche) and incubated at RT for 30 min. The lysate was clarified by centrifugation at 6000× *g* for 20 min. at 4 °C and subsequently added to Ni-NTA Magnetic Agarose Beads (Qiagen). After extensive washing with buffer containing 20 mM HEPES pH 8.0, 500 mM NaCl, 20 mM imidazole, 1 mM TCEP, and 0.05% (v/v) Tween-20, the protein was eluted in buffer containing 20 mM HEPES pH 8.0, 500 mM NaCl, 250 mM imidazole, 1 mM TCEP, and 0.05% (v/v) Tween-20. The eluted protein was analyzed on NuPAGE 4–12% Bis-Tris polyacrylamide gels (Thermo Fisher Scientific) followed by InstantBlue Coomassie staining (Abcam).

### CRBN$^{midi}$ expression and purification
The His$_6$-tagged fusion protein was expressed in BL21(DE3) *E. coli* cells grown in LB supplemented with 50 µg/mL kanamycin. Protein expression was induced with 0.5 mM IPTG at 18 °C overnight with 50 µM ZnCl$_2$ added at induction. After harvesting, the cells were resuspended in buffer containing 20 mM HEPES pH 8.0, 500 mM NaCl,

50 µM ZnCl$_2$, 0.5 mM TCEP, 0.05% (v/v) Tween-20, 5 mM imidazole, 1 mM MgCl$_2$, cOmplete Protease Inhibitor (Roche), and 10 µg/mL DNase I and lysed using a Continuous Flow Cell Disruptor (Constant Systems) at 30,000 psi. The lysate was clarified by centrifugation at 48,000 × g for 30 min. at 4 °C and loaded onto a HisTrap HP (Cytiva) column. Impurities were washed with buffer containing 20 mM HEPES pH 8.0, 500 mM NaCl, 0.5 mM TCEP, and 20 mM followed by 60 mM imidazole. The protein was eluted by increasing the imidazole concentration to 100–125 mM. The eluted protein was loaded onto a HiPrep Sephadex G-25 Desalting Column pre-equilibrated with 20 mM HEPES pH 8.0, 500 mM NaCl, 0.5 mM TCEP to remove imidazole. The protein was incubated overnight at 4 °C with His$_6$-tagged TEV protease. The TEV protease, uncleaved protein, His$_6$ tag, and remaining impurities were removed by reverse Ni$^{2+}$ affinity chromatography using HisTrap HP columns with CRBN$^{midi}$ eluting in buffer containing 10–20 mM imidazole, 20 mM HEPES pH 8.0, 500 mM NaCl, and 0.5 mM TCEP. The protein-containing fractions were concentrated (Amicon Ultra centrifugal filter, MWCO 10 kDa, Merck) and further purified by size-exclusion chromatography (SEC) on a Superdex 200 column (Cytiva) in buffer containing 20 mM HEPES pH 7.5, 500 mM NaCl, and 0.5 mM TCEP. The protein was concentrated to around 3 mg/mL, flash frozen in liquid nitrogen and stored at -80 °C until further use. All chromatography purification steps were performed using an Äkta Pure (Cytiva) system at 4 or 20 °C.

### CRBN$^{TBD}$ expression and purification
The glutathione-S-transferase (GST) fusion protein was expressed in BL21(DE3) *E. coli* cells grown in LB supplemented with 50 µg/mL kanamycin. Protein expression was induced with 0.5 mM IPTG at 18 °C overnight with 50 µM ZnCl$_2$ added at induction. After harvesting, the cells were resuspended in buffer containing 50 mM HEPES pH 8.0, 500 mM NaCl, 1 mM TCEP, 1 mM MgCl$_2$, cOmplete Protease Inhibitor (Roche), and 10 µg/mL DNase I (Roche) and lysed using Continuous Flow Cell Disruptor (Constant Systems) at 30,000 × psi. The lysate was clarified by centrifugation at 40,000 × g for 40 min. at 4 °C and added to 7 mL of glutathione agarose beads equilibrated in 50 mM HEPES pH 8.0, 500 mM NaCl and 1 mM TCEP. The lysate was incubated for 2 hours at 4 °C before it was applied to a batch column and the impurities were washed with 50 mM HEPES pH 8.0, 500 mM NaCl and 1 mM TCEP. The protein was incubated overnight at 4 °C with His$_6$-tagged TEV protease for on-bead cleavage of the GST tag. The cleaved protein was eluted by washing the beads with 50 mM HEPES pH 8.0, 500 mM NaCl and 1 mM TCEP and collecting the flow through. The protein was concentrated (Amicon Ultra centrifugal filter, MWCO 3 kDa, Merck) and further purified by SEC on a Superdex 75 column (Cytiva) in buffer containing 20 mM HEPES pH 7.5, 200 mM NaCl, and 1 mM TCEP. The protein was concentrated to around 20 mg/mL, flash frozen in liquid nitrogen and stored at -80 °C until further use. All chromatography purification steps were performed using an Äkta Pure (Cytiva) system.

### BRD4$^{BD2}$ expression and purification
Human Brd4$^{BD2}$ (residues 333–460, UniProt ID: O60885) was expressed with an N-terminal His$_6$ tag in *E. coli* BL21(DE3) at 37 °C with LB supplemented with 50 µg/mL kanamycin. Once OD$_{600}$ reached 0.8, protein expression was induced with 0.5 mM IPTG, followed by addition of 50 µM ZnCl$_2$, and grown overnight at 18 °C. After harvesting, the cells were resuspended in buffer containing 12 mM phosphate buffer, 500 mM NaCl, 40 mM imidazole, 1 mM MgCl$_2$ and DNase I (10 µg/mL), and cells were lysed using a Continuous Flow Cell Disruptor (Constant Systems) at 30,000 psi. Cell lysates were clarified by centrifugation at 18,000 × g for 30 minutes at 4 °C. The lysate was filtered and loaded onto a HisTrap FF affinity column (GE Healthcare) and eluted with 12 mM phosphate buffer, 500 mM NaCl and 500 mM imidazole. The final purified proteins were concentrated to around 9 mg/mL and

stored in 20 mM HEPES pH 7.5, 500 mM NaCl and 0.5 mM TCEP at -80 °C until further use. All chromatography purification steps were performed using a BioRad NGC system at 4 °C.

### IKZF1$^{ZF2}$ expression and purification
IKZF1$^{ZF2}$ (residues 141–174, UniProt ID: Q13422), which was incorporated into a pGEX6P-1 vector, was expressed in *E. coli* BL21(DE3) using LB supplemented with 50 µM ZnSO$_4$ and 100 µg/mL ampicillin. Protein expression was induced using 0.5 mM IPTG when OD$_{600}$ reached 0.6 - 0.8. The cells were harvested through centrifugation and subsequently re-suspended in a buffer comprising 20 mM Tris-HCl pH 8.0, 500 mM NaCl, 5% (v/v) glycerol, and 0.1 mM dithiothreitol (DTT). The protein purification was performed as previously described[44] with gel filtration conducted using a buffer composed of 50 mM HEPES pH 7.5, 150 mM NaCl, and 0.25 mM TCEP.

### Crystallization of apo CRBN$^{midi}$
Crystals of apo CRBN$^{midi}$ were grown using the sitting drop vapour diffusion method at 20 °C by mixing equal volumes of protein solution (2.9 mg/mL in SEC buffer) with reservoir solution containing 0.2 M sodium citrate, 20% (w/v) PEG 3350, and 0.1 M BIS-TRIS propane pH 6.5. The crystal was cryoprotected in the reservoir solution supplemented with 20% (v/v) glycerol and subsequently flash cooled in liquid nitrogen.

Diffraction data were collected at Diamond Light Source beamline I24 using a Pilatus3 6 M detector at 100 K and 1.00 Å wavelength. The data was processed using autoPROC (version 1.0.5)[52] and indexed in space group C222$_1$. The structure was solved by molecular replacement in phenix.phaser (version 1.20.1-4487)[53] using the atomic coordinates of the mezigdomide-bound CRBN$^{midi}$ structure as search model, assuming one molecule in the asymmetric unit. Refinement and model building was done iteratively using phenix.refine (version 1.20.1-4487)[54] and coot (WinCoot 0.9.8.1)[55]. The Ramachandran statistics for the final structure were: 97.5% favoured, 2.5% allowed and 0% outliers.

### Crystallization of CRBN$^{midi}$ bound to mezigdomide
Crystals of CRBN$^{midi}$ bound to mezigdomide were grown using the sitting drop vapour diffusion method at 4 °C by mixing equal volumes of protein solution (4 mg/mL) supplemented with 214 µM mezigdomide in buffer containing 20 mM HEPES pH 7.5, 500 mM NaCl, 0.5 mM TCEP, 1% (v/v) DMSO with reservoir solution containing 0.2 M sodium acetate, 25% (w/v) PEG 3350, and 0.1 M BIS-TRIS pH 6.5. The crystals were cryoprotected in the reservoir solution supplemented with 20% (v/v) ethylene glycol and subsequently flash cooled in liquid nitrogen.

Diffraction data were collected at Diamond Light Source beamline I24 using a Pilatus3 6 M detector at 100 K and 1.00 Å wavelength. The data was processed using autoPROC (version 1.0.5)[52] and STARANISO (version 2.3.79)[56], indexed in space group P4$_3$2$_1$2. The structure was solved by molecular replacement in phenix.phaser[53] using a model predicted by ColabFold (version 1.2)[57] for the CRBN$^{midi}$ sequence as a search model, assuming one molecule in the asymmetric unit. Refinement and model building was done iteratively using phenix.refine[54] and coot (WinCoot 0.9.8.1)[55]. The Ramachandran statistics for the final structure were: 95.8% favoured, 4.2% allowed and 0% outliers.

### Crystallization of CRBN$^{midi}$ bound to lenalidomide
CRBN$^{midi}$ at 3 mg/mL in a buffer containing 20 mM HEPES pH 7.5, 500 mM NaCl and 0.5 mM TCEP was mixed with a 4-molar excess of lenalidomide (320 µM, 1.7% (v/v) DMSO final). The complex was subjected to co-crystallization using the sitting drop vapour diffusion method across several sparse matrix screens at 20 °C by mixing equal volumes of protein solution and reservoir solution. Initial crystal hits from ProPlex well C5 (0.1 M Tris pH 8, 20% (w/v) PEG 4000) were optimized using the Hampton Research additive screen and the best diffracting crystals resulted from a reservoir solution containing 0.1 M

Tris pH 8, 20% (w/v) PEG 4000 and 1.2% myo-inositol. The crystals were harvested directly from the drop and flashed cooled in liquid nitrogen.

Diffraction data were collected at Diamond Light Source beamline i24 using the CdTe Eiger2 9 M detector at a wavelength of 0.6199 Å. The 2.5 Å dataset was processed using autoPROC (version 1.0.5)[52], indexed in $C222_1$ space group and the structure was solved by molecular replacement in phenix.phaser[53] using the CRBN$^{midi}$:mezigdomide coordinates as a search model. Phenix.refine[54] and coot[55] were used for refinement and model building. The Ramachandran statistics for the final structure were: 93.29% favored, 6.71% allowed and 0% outliers.

## Crystallization of CRBN$^{midi}$ bound to compound 1

CBRN$^{midi}$ at 3.4 mg/mL in a buffer containing 20 mM HEPES pH 7.5, 500 mM NaCl and 0.5 mM TCEP was mixed with a 4-molar excess of a racemic mixture of compound **1** (364 μM, 1.8% (v/v) DMSO final). The complex was subjected to co-crystallization using the sitting drop vapour diffusion method across several sparse matrix screens at 20 °C by mixing equal volumes of protein solution and reservoir solution. Initial crystal hits from Morpheus F2 (Molecular Dimensions) were optimized and the best diffracting crystals were grown in 100 mM Buffer system I (Molecular Dimensions), 15.45% (v/v) Precipitant mix 2 (MD), 180 mM Monosaccharides (MD). The crystals were harvested directly from the drop and flash-cooled in liquid nitrogen.

Diffraction data were collected at Diamond Light Source beamline I04 using the Eiger2 XE 16 M detector at a wavelength of 0.9537 Å. The 1.95 Å dataset was processed using autoPROC (version 1.0.5)[52], indexed in $P2_12_12_1$ space group and the structure was solved by molecular replacement in phenix.phaser[53] using the CRBN$^{midi}$:lenalidomide coordinates as a search model. Phenix.refine[54] and coot[55] were used for refinement and model building. Ligand restraints were generated for compound **1** using SMILES with undefined stereochemistry and (R)-compound **1** was modelled in the structure as it best fit the electron density. The Ramachandran statistics for the final structure were: 96.63% favored, 3.21% allowed and 0.16% outliers.

## Crystallization of a CRBN$^{midi}$:CFT-1297:BRD4$^{BD2}$ ternary complex

CRBN$^{midi}$, CFT-1297 and Brd4$^{BD2}$ were mixed at a 1:4:4 ratio and purified as a ternary complex by SEC using a Superdex-75 10/300 Increase GL column (GE Healthcare). The ternary complex-containing fractions were pooled and concentrated to a final concentration of 3.5 mg/mL. The ternary complex was crystallized using the sitting-drop vapour diffusion method by mixing equal volumes of protein with 20% (w/v) PEG 3350, 0.2 M sodium malonate and 0.1 M Bis-Tris Propane pH 6.5. Small crystals, which appeared after one week were used as a seeding stock to optimize the crystal size. The seed was diluted with 300 μL of the crystallization condition. The final crystals were grown by mixing the ternary complex, seed stock, and a mixture of 20% (w/v) PEG 3350, 0.2 M sodium citrate and 0.1 M Bis-Tris Propane pH 7.5 with a ratio of 1:0.1:0.9. Crystals appeared in one day and grew to their full size in one week at 20 °C. The crystals were harvested and flash-cooled in liquid nitrogen using 30% (v/v) glycerol in the crystallization solution as a cryo-protectant.

Diffraction data were collected at Diamond Light Source beamline I24 using the Eiger CdTe 9 M detector at a wavelength of 0.6199 Å. Indexing and integration of reflections were performed using xia2.dials (version 3.14.1), and scaling and merging with xia2.multiplex (version 3.14.0)[58]. The space group was determined to be P1. The structure was solved by molecular replacement in phenix.phaser[53] using CRBN$^{midi}$:mezigdomide and Brd4$^{BD2}$ (PDB ID: 2OUO) as search models. Two instances of the ternary complex were found in the asymmetric unit, indicating a final solvent content of 50.09% as calculated from the Matthews coefficient. The initial model was refined iteratively using coot (WinCoot 0.9.8.1)[55] and phenix.refine[54]. Ligand structures and restraints were generated using the PRODRG server[59].

The Ramachandran statistics for the final structure were: 96.6% favoured, 3.4% allowed and 0% outliers.

## Crystallization of a CRBN$^{midi}$:mezigdomide:IKZF1$^{ZF2}$ ternary complex

CRBN$^{midi}$, mezigdomide and IKZF1$^{ZF2}$ were mixed in a 1:4:4 ratio. The complex was crystallized using the sitting drop vapour diffusion method by mixing equal volumes of the ternary complex (107 μM) and reservoir solution containing 25% (w/v) PEG 3350, 0.2 M ammonium sulphate and 0.1 M HEPES pH 7.5. The crystal was cryoprotected in the reservoir solution containing 30% (v/v) ethylene glycol and flash cooled in liquid nitrogen.

Diffraction data collection was carried out at Diamond Light source beamline I24 using an Eiger CdTe 9 M detector at a wavelength of 0.9537 Å. The data was processed using xia2.dials[58]. The crystal belonged to the space group $P12_11$. The structure was solved by molecular replacement in phenix.phaser[53] using chain A from CRBN$^{midi}$:CFT-1297:BRD4$^{BD2}$ and chain L from the CRBN:DDB1:pomalidomide:IKZF$^{ZF1}$ complex (PDB ID: 6H0F) as search models. The initial model was refined iteratively using coot (WinCoot 0.9.8.1)[55] and phenix.refine[54]. The Ramachandran statistics for the final structure were: 94.3% favoured, 5.7% allowed and 0% outliers.

### Analysis of crystal structures

The RMSD values for all structural alignments were calculated using the super command over Cα atoms in PyMOL (version 4.6, Schrödinger). Structure figures were generated in PyMOL (version 4.6, Schrödinger).

### SAXS

SAXS data were collected at beamline B21 at the Diamond Light Source in SEC-SAXS configuration (March 2023 and June 2024)[60]. CRBN$^{midi}$ was concentrated to 6.5 mg/mL for apo, mezigdomide and lenalidomide binding measurements or 3.4 mg/mL for Boc-VcN and Boc-AcQ binding measurements in SAXS buffer (20 mM HEPES pH 7.5, 500 mM NaCl, 0.5 mM TCEP) immediately before the experiment. Binary complexes were formed by addition of binders to 400 μM (mezigdomide and lenalidomide) or 200 μM (Boc-VcN and Boc-AcQ) final binder concentration in 4% DMSO and incubation on ice for 30 minutes. Samples were run on a Superdex S200 3.2/300 column (Cytiva) in SAXS buffer with 0.075 mL/min. flowrate at 298 K coupled online to the SAXS beamline. SAXS data were acquired using a 3 s exposure at 12.4 kV and 3.6 m detector (Eiger 4 M) distance across a q-range of $4.5 \times 10^{-3}$ - $3.4 \times 10^{-1}$ Å$^{-1}$. Frames from SEC-SAXS that corresponded to the main protein peak, were merged using Chromixs[61] and analyzed using the ATSAS (version 3.2.1) package[62]. Radii of gyration and molecular mass were extrapolated from the Guinier plot using Primus. Pairwise-distance distributions were calculated using GNOM[63]. The dimensionless Kratky plot[49] was obtained by manually plotting $(qR_g)^2 I(q)/I_0$ against $qR_g$.

The scattering curves were compared to scattering curves backcalculated from the crystal structure of CRBN$^{midi}$:mezigdomide (PDB ID: 8RQ8) in the closed conformation and a homology model of the open conformation using CRYSOL[64]. The open conformation model was obtained by homology modelling using the SWISS-MODEL webserver (accessed April 18$^{th}$ 2023)[65] with the cryo-EM structure of apo-CRBN (PDB ID: 8CVP)[25] as the input structure.

### Surface plasmon resonance

All SPR measurements were carried out on a Biacore™ 8 K (Cytiva, Biacore Insight Control Software version 5.0.18.22405). CRBN$^{midi}$ was immobilized via a His$_6$ tag on a NiHC 1500 M chip (XanTec) at 25°C. The chip surface was prepared for ligand (CRBN$^{midi}$) capture as per manufacturer's instructions, flowing 350 mM EDTA, followed by buffer, then loaded using 5 mM NiCl$_2$ followed by buffer. For binary measurements CRBN$^{midi}$ was diluted to 350 nM and flowed at 10 μL/min. for 420 s to

capture a final bound response of 240 RUs. For binary measurements, 10 nM of CRBN$^{midi}$ was flowed at 10 μl/min. for 420 s to capture a final bound response of approximately 10,000 RUs. The buffer for immobilization and all kinetic measurements was 50 mM Tris pH 8.0, 150 mM NaCl, 0.25 mM TCEP, 0.005% (v/v) Tween-20 and 2% (v/v) DMSO.

Kinetic measurements were derived from a multicycle kinetics measurement, with a 5-point 3-fold dilution series of analyte CFT-1297 (5 μM to 20.6 nM). For ternary experiments the analyte was supplemented with 50 μM BRD4$^{BD2}$. Each concentration point was flowed independently at 50 μL/min. with an association time of 150 s followed by 500 s dissociation. All measurements were carried out at 20 °C.

Data was analyzed using Biacore™ Insight Evaluation Software (version 4.0.8.20368). Raw sensograms were solvent corrected, followed by reference and blank subtraction. To calculate $K_D$ values, data was globally fitted to 1:1 binding model. Alpha, a parameter indicative of cooperativity, was calculated by dividing binary $K_D$ by ternary $K_D$.

### TR-FRET

A half-log dilution series of CFT-1297 was incubated with His$_6$-tagged CRBN constructs and Cy5-labelled BRD4$^{BD2}$ in TR-FRET buffer (50 mM HEPES pH 7.5, 150 mM NaCl, 0.5 mM TCEP, 0.01% (v/v) Tween-20, 2% (v/v) DMSO) for 30 minutes before addition of α-his Eu donor beads (20 ug/mL, Perking Elmer). The mixture was incubated in the dark with gentle rocking for 1 hour before readout using a PHERAstar microplate reader (BMG Labtech) with Pherastar software (version 5.70 R5). All incubations were conducted at 20 °C. Data were fitted by non-linear regression (curve fit) using Prism software (GraphPad version 10). Experiments were conducted in triplicate and were the average of three repeats.

### Differential scanning fluorimetry

Samples for the experiment shown in Supplementary Fig. 4 were prepared in a total volume of 25 μL by mixing 4.8 μM CRBN$^{midi}$ or 3.2 μM CRBN$^{Δ40}$:DDB1$^{ΔBPB}$ (purchased from Selvita) with SYPRO orange (Merck, S5692) at 5× final concentration and 100 μM lenalidomide when applicable in a final buffer containing 50 mM HEPES pH 7.5, 200 mM NaCl, 0.5 mM TCEP and 4% (v/v) DMSO. The temperature was increased from 25 to 95 °C at a rate of 1 °C/min. using a CFX96 Real-Time C1000 Touch Thermal Cycler (Biorad) instrument. The melting temperature was calculated as the average of the minima of the derivative (-dRFU/d$T$) of two replicate samples. The melting curves were plotted using Prism software (GraphPad version 10).

Samples for the experiment shown in Fig. 2f were prepared in a total volume of 15 μL by mixing 10 μM of CRBN$^{midi}$ with 100 μM of compound when applicable in a final buffer containing 50 mM HEPES pH 7.5, 200 mM NaCl, 0.5 mM TCEP and 4% (v/v) DMSO. Samples were loaded into Prometheus standard capillaries (NanoTemper Technologies) and thermal unfolding experiments were carried out by increasing the temperature from 20 °C to 95 °C at a rate of 1 °C/min. using a Prometheus Panta (NanoTemper Technologies). Protein aggregation was assessed, and turbidity values were calculated by the PR.Panta control software (NanoTemper Technologies). First derivative curves were plotted using Prism software (GraphPad version 10).

### Isothermal titration calorimetry

Experiments were carried out on an ITC200 instrument (Malvern), using MicroCal ITC200 (version 1.26.1) software, in buffer containing 50 mM HEPES pH 7.5, 500 mM NaCl, 1 mM TCEP, 4% (v/v) DMSO at 298 K stirring the sample at 600 rpm. The ITC titration consisted of a 0.4 μL initial injection (discarded during data analysis) followed by 19 × 2 μL injections with 180 s spacing between injections. Lenalidomide (400 μM) was directly titrated into CRBN$^{midi}$ (20 μM). For a control titration, 400 μM lenalidomide was titrated into buffer. The data was fitted using a one-set-of-site binding model to obtain dissociation constants, binding enthalpy (ΔH), and stoichiometry (N) using Micro-Cal PEAQ-ITC Analysis Software (version 1.1.0.1262, Malvern).

### Molecular dynamics (MD)

The CRBN$^{midi}$:mezigdomide crystal structure was pre-treated using the Protein Preparation Workflow (PPW)[66] available in Schrödinger Suite (Release 2022-4) before initiating the MD simulation experiments. The PPW treatment involved preparing the structure at pH 7.4, by adding hydrogen atoms, capping of the N and C termini with ACE (N-acetyl) and NMA (N-methyl amide) groups, respectively, and building the missing side chain atoms. The missing residues were built using Prime[67]. The hydrogen bond network was optimized by reorienting hydroxyl and thiol groups, water molecules, amide groups of asparagine (Asn) and glutamine (Gln), and the imidazole ring in histidine (His); and predicting protonation states of histidine, aspartic acid (Asp) and glutamic acid (Glu) and tautomeric states of histidine. The resulting structure was minimized to relieve any strain and fine-tune the placement of various groups. Hydrogen atoms were optimized fully, which allows relaxation of the H-bond network. Heavy atoms were restrained, so that a small amount of relaxation is allowed (RMSD = 0.30 Å). Mezigdomide-bound CRBN$^{ΔN}$ structure as obtained from the Protein Data Bank (PDB ID: 8D7U) was also subjected to identical pre-treatment with the PPW. The final structures from the PPW were used as input for MD simulations. The DDB1 protein chain in the structure 8D7U was not considered for the MD simulations of the mezigdomide-bound CRBN$^{ΔN}$ structure. OPLS4 force field[68] was used throughout the computational studies.

MD simulations were carried out using the Desmond Molecular Dynamics System[69] implemented within the Schrödinger Suite Release 2022-4 and 2023-1 (Schrödinger, LLC). The systems for MD simulations were built using identical protocols with the input structures (CRBN$^{midi}$:mezigdomide complex and CRBN$^{ΔN}$:mezigdomide complex) being placed in an orthorhombic box (with buffer distance of 15 Å) of explicit water molecules (TIP3P water model). A salt concentration of 0.15 M was maintained and ions (Na$^+$, Cl$^-$) were placed to neutralize the systems. Supplementary Table 5 summarizes the system setup parameters.

The built systems were relaxed before the production run to get rid of any strains. The relaxation process involved a series of minimizations and short molecular dynamics simulations. The details of the various stages of the relaxation protocol are available in the supplementary note. NPT ensemble class (pressure = 1 bar; temperature = 298.15 K) was used. The relaxed systems were subjected to 100 ns[70] of MD simulation production runs with a recording interval of 10 ps for trajectory and energy, resulting in 10,000 frames per simulation. RESPA integrator with 2 fs for bonded and near and 6 fs for far time-steps was used. Noose-Hoover chain thermostat (relaxation time = 1 ps) and Martyna-Tobias-Klein barostat (relaxation time = 2 ps) methods were used. Coulombic interactions were computed using the *u-series*[71] decomposition method with a cut-off radius of 9 Å. For each system, we performed five different simulations of 100 ns each production run with random seeds and randomizing the velocities at the beginning of the calculations. The Root Mean Square Deviations (RMSD, measured in Å) of all heavy protein atoms (Supplementary Fig. 6) were calculated for each run (5 × 100 ns = 500 ns) of both systems. The Root Mean Square Fluctuations (RMSF, measured in Å) of all heavy atoms in protein residues as derived from the MD simulations and reported in this study are the averages over the five independent runs. The protein residues that are involved in various types of noncovalent interactions with the ligand atoms for more than 20% of the entire simulation time (100 ns) in each run were noted to compare the interaction profiles (interaction types and their stabilities) of mezigdomide bound to the two different protein constructs—CRBN$^{midi}$ and CRBN$^{ΔN}$. The input system as well as the first and the last frame from each run of the simulations are provided as.pdb files (see Supplementary Data 1). All

observations reported in this study are based on the data obtained from the production runs.

## Reporting summary

Further information on research design is available in the Nature Portfolio Reporting Summary linked to this article.

## Data availability

X-ray crystallography data generated in this study have been deposited in the Protein Data Bank (PDB) under accession codes: 8RQ1 (apo CRBN$^{midi}$), 8RQ8 (CRBN$^{midi}$:mezigdomide), 8RQ9 (CRBN$^{midi}$:CFT-1297:BRD4$^{BD2}$), 8RQA (CRBN$^{midi}$:lenalidomide), 8RQC (CRBN$^{midi}$:mezigdomide:IKZF1$^{ZF2}$), 9GAO (CRBN$^{midi}$:compound 1). The SAXS data have been deposited to Small Angle Scattering Biological Data Bank (SASBDB) under accession numbers: SASDU52 (apo CRBN$^{midi}$), SASDU62 (CRBN$^{midi}$:mezigdomide), SASDU92 (CRBN$^{midi}$:lenalidomide), SASDVN6 (CRBN$^{midi}$:Boc-VcN), SASDVP6 (CRBN$^{midi}$:Boc-AcQ). The His$_6$-CRBN$^{midi}$ expression plasmid has been deposited to Addgene under accession number #215330. Source data are provided with this paper.

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

## Acknowledgements

We thank Adam Bond for the gift of BRD4[BD2] protein, Diane Cassidy for technical support, Thomas Eadsforth (Drug Discovery Unit, University of Dundee) for the gift of the His$_6$-MBP-TEV-pNIC28-Bsa4 vector, and Moriz Mayer (Boehringer Ingelheim) for performing confirmatory NMR spectra with compound CFT-1297. We thank Boehringer Ingelheim RCV GmbH & Co KG for very kindly providing us with the compounds Boc-VcN and Boc-AcQ. We acknowledge the Diamond Light Source (proposals mx26793, mx35324, sm33832 and sm38813) for provision of synchrotron radiation facilities, and we would like to thank staff of beamlines I04, I24, and B21 for assistance and support with MX and SAXS data collection. This work was funded by the pharmaceutical companies (Almirall, Boehringer Ingelheim, Eisai and Tocris-Biotechne) who are providing sponsored research funding to the AC laboratory. Funding is also gratefully acknowledged from the Innovative Medicines Initiative 2 (IMI2) Joint Undertaking under grant agreement no. 875510 (EUbOPEN project). The IMI2 Joint Undertaking receives support from the European Union's Horizon 2020 research and innovation program, European Federation of Pharmaceutical Industries and Associations (EFPIA) companies, and associated partners KTH, OICR, Diamond, and McGill. HF received funding from a Japan Society for the Promotion of Science (JSPS) Postdoctoral Fellowship, no. 23KJ1669.

## Author contributions

Conceptualization: S.R., D.Z., A.C. Methodology: A.K., V.A.S., S.R., S.C., W.F., D.Z., A.C. Investigation: A.K., V.A.S., H.F., D.D., S.R., Z.R., S.C., K.H., J.P., D.G., A.W., M.R.R., M.S., D.M.L., M.N., W.F., D.Z. Visualization: A.K., V.A.S., H.F., S.R., S.C., K.H., D.Z., A.C. Funding acquisition: A.C. Supervision: S.R., W.F., D.Z., A.C. Writing—original draft: A.K., V.A.S., H.F., D.D., Z.R., S.C., K.H., D.Z. Writing—review & editing: A.K., V.A.S., W.F., D.Z., A.C.

## Competing interests

The Ciulli laboratory receives or has received sponsored research support from Almirall, Amgen, Amphista Therapeutics, Boehringer Ingelheim, Eisai, Merck KaaG, Nurix Therapeutics, Ono Pharmaceutical and Tocris-Biotechne. AC is a scientific founder and shareholder of Amphista Therapeutics, a company that is developing targeted protein degradation therapeutic platforms. The remaining authors declare no competing interests.
