## [Peer Review File · Nature Communications]

Design of a Cereblon construct for crystallographic and biophysical studies of protein degradersREVIEWER COMMENTS

Reviewer #1 (Remarks to the Author):

I was requested to comment on the SAXS and MD simulations reported in the manuscript “Design of a Cereblon construct for crystallographic and biophysical studies of protein degraders.” Overall, the application of SAXS and MD simulations support the conclusions of the study. I have only minor comments:

Minor comments:

1. When describing the rearrangement of the TBD and Lon domains between open and closed state the authors provide a reference to Petzold et al. [1]. Maybe I missed it, however, I did not find an open structure in the given reference. Please clarify.
2. The authors report that in absence of a small-molecule ligand, CRBNmidi adopted the closed conformation. This contradicts the findings by Watson et al. that apo-CRBN always adopts an open conformation [2]. Could the authors elaborate on the reasons why apo-CRBNmidi adopts the closed conformation within the crystal? Watson et al. suggest that interactions of the sensor loop with a helix in the HB domain (residues ~210 to 220) and with a loop in BPC in DDB1 (residues ~776 to 780) play a role in maintaining the open conformation of CRBN [2]. Could the authors comment on the role of the sensor loop in the apo-CRBNmidi crystal?
3. The authors compared their measured SAXS intensities to back-calculated curves from the crystal structure of CRBNmidi:mezigdomide in closed conformation and a homology model of the open conformation using CRY SOL. Notably, while using the same model in the first and third graph and second and fourth graph, respectively, the back-calculated curves are different. This is possibly the result of the large number of fitting parameters used by CRY SOL (for the hydration shell, two for the excluded solvent, optionally a constant offset). The authors may consider using a method that uses explicit solvent and thus does not require free fitting parameters for back-calculating a SAXS curve, such as WAXSiS [3,4] or CAPRIQORN [5].
4. In the section on Molecular Dynamics some details are missing:
 - a. Please provide details how the structure 8D7U was modified, e.g. that DDB1 was removed.
 - b. Please explain over which atoms the RMSF was calculated, such as C α , backbone or heavy atoms
 - c. Please provide details on how the Protein-ligand interaction stability were calculated.
 - d. The authors report on the 9Å cut-off radius that was set for the Coulomb interactions. Plain cut-offs for Coulomb interactions are uncommon since 25 years because they may lead to severe artifacts. Was really a plain cut-off used, and not, for instance, particle-mesh Ewald (PME)? Do the authors possibly refer to the direct-space cutoff within the PME method?
5. In Figure 1 A there is a square in the superscript, which likely should be a Δ to report on CRBN Δ N.
6. I suggest adding a reference to figure 1 D when discussing the superposition of CRBNmidi with the corresponding portion of PDB ID 4TZ4.

7. Please add units to the RMSF in figure S3 A, B, C, D.

8. I suggest adding a line legend to figure 3 or describing the lines in the caption.

[1] G. Petzold, E.S. Fischer & N.H. Thomä, Structural basis of lenalidomide-induced CK1 α degradation by the CRL4CRBN ubiquitin ligase. *Nature* 532, 127-130 nature16979 (2016).

[2] E.R. Watson et al. Molecular glue CELMoD compounds are regulators of cereblon conformation. *Science* 378, 549-553 (2022)

[3] P. Chen and J.S. Hub, Validating solution ensembles from molecular dynamics simulations by wide-angle X-ray scattering data, *Biophys. J.*, 107, 435-447 (2014)

[4] C.J. Knight, J.S. Hub, WAXSiS: a web server for the calculation of SAXS/WAXS curves based on explicit-solvent molecular dynamics. *Nucleic Acids Res.* 43(W1), W225–W230 (2015)

[5] J. Köfinger, G. Hummer, Atomic-resolution structural information from scattering experiments on macro- molecules in solution. *Phys. Rev. E* 87(5), 052712 (2013)

Reviewer #2 (Remarks to the Author):

Although CRBN has been developed as a major E3 ligase handle for degrader molecules, there is a relative paucity in structural data for ternary complexes with neo-substrates. Some biophysical studies are also impeded by challenges in producing quality protein due to both CRBN's intrinsic conformational heterogeneity and a requirement for coexpression with DDB1. The present work provides a new system for facile production of a "bonsai", or miniaturized version of CRBN. This CRBNmidi retains the two domains that are crucial for binding to degrader molecules and substrates (the Lon domain and thalidomide binding domain, TBD). The authors achieved their goal after using standard protein engineering approaches of deleting the flexible region, domains that are unnecessary for ternary complex formation, introducing solubilizing mutations in place of "sticky" hydrophobic surfaces, and a C/S mutation known to improve crystallization. The culmination of these efforts yielded 1.5 mg/liter of pure, CRBNmidi using straightforward approaches for expression in *E. coli* and purification.

The authors pressure test CRBNmidi for recapitulation of native CRBN features and utility in a battery of assays used for developing degrader molecules. They performed SACS analysis that suggested that degrader-bound CRBNmidi has a smaller radius of gyration than the apo protein. This idea is also supported by MD simulations. They show CRBNmidi performs well in biophysical methods of TR-FRET, SPR, and ITC.

The authors solved crystal structures of CRBNmidi in complexes with lenalidomide and mezigdomide:IKZF1ZF2 that superimpose well with the known structures. They solve a structure of CRBNmidi:mezigdomide that reveal an additional interaction with the ligand fluorine. And they obtained a new structure: CRBNmidi:PROTAC CFT-1297:BRD4BD2. This had two molecules in the asymmetric unit that slightly differ in relative interaction with the neo-substrate.

Overall, the study reports a new reagent, CRBNmidi, that promises substantially reduce the effort and cost for biophysical and structural analysis of ligand and ternary complex interactions with the most predominant E3 ligase used in targeted protein degradation. This will democratize the field by making such studies broadly accessible, and could greatly advance development of degrader reagents and ideally therapeutics. As such, I recommend publication in Nature Communications after addressing the comments below.

Main comments:

The authors' own findings that CRBNmidi:PROTAC CFT-1297:BRD4BD2 crystallizes in two orientation highlights an additional potential challenge to obtaining CRBN ternary complex structures: neosubstrate binding could be heterogeneous. Crystallography can aid in obtaining structures of such complexes through crystal packing but such interactions often limit resolution by cryo-EM.

1. This point should be raised in the discussion.

2. The authors should show the crystal packing interactions to reveal the extent to which interactions in the binary and ternary complexes are stabilized by crystal packing. If the lattice constrains positions available to the neo substrate, this should be described in the results and discussion. The discussion could frame this as both a strength and a weakness.

Reviewer #3 (Remarks to the Author):

Reviewer #4 (Remarks to the Author):

The manuscript presents CRBNmidi, a new cereblon (CRBN) construct engineered for high-yield expression from *E. coli*. The ubiquitin E3 ligase CRBN, utilized extensively in targeted protein degradation, serves as a recruitment platform for PROTACs and molecular glue degraders. However, structure-guided drug design and in vitro biophysical characterization of CRBN-based degraders were partially impeded by difficulties in producing a suitable recombinant CRBN. CRBNmidi, which retains the Lon and thalidomide binding domains but excludes the DDB1 binding domain, addresses this issue by enhancing solubility and stability without the adaptor protein DDB1, avoiding the self-aggregation typically observed when CRBN is expressed alone. Through high-resolution X-ray crystallography and cryoEM of CRBNmidi with degrader molecules such as lenalidomide, mezigdomide, and CFT-1297, the authors have elucidated detailed protein-ligand and protein-protein interactions previously unobserved with other CRBN constructs. Additionally, the suitability of CRBNmidi for in vitro assays has been confirmed through various biophysical ligand-binding studies, including SAXS, DSF, ITC, TR-FRET, and SPR.

CRBNmidi appears to be a promising platform for high-resolution structural studies and the design of new degraders. This work will likely interest those in the protein degradation and drug discovery communities. However, the utility and applicability of CRBNmidi in the context of CRBN's

physiological functions and as a tool to study native interactions with its endogenous substrates remain uncertain.

I would support the publication of this manuscript in Nature Communications if the following points are addressed:

1. CRBNmidi lacks certain domains and includes 12 site mutations, so its interaction dynamics might differ from the full-length native CRBN, potentially leading to overlooking some physiological interactions and allosteric effects. The authors should assess the physiological relevance of CRBNmidi by investigating whether it adequately models native CRBN functions. The authors should verify whether CRBNmidi maintains a consistent binding mode with at least one or two known native CRBN substrates (e.g., MEIS2, APP, GLUL) and/or engineered/endogenous proteins with C-terminal cyclic imide degrons.

2. The manuscript's claim that CRBNmidi's readiness to crystallize in a ligand-bound form is an improvement over previous constructs should be carefully rephrased to acknowledge its limitations. Specifically, the limited utility of CRBNmidi for designing allosteric binders of CRBN or small molecules that influence the open vs. closed conformation of CRBN upon binding should be clearly stated.

3. The engineered mutations and deletions might artificially stabilize specific conformations, as evidenced by apo CRBNmidi adopting a closed conformation without ligands, which could bias structural and functional outcomes and may render CRBNmidi unresponsive to binders with weaker affinities than well-characterized thalidomide derivatives. Since molecular glues with weaker binding affinities are beneficial due to their minimal inhibitory effects on E3 ligase interactions with native substrates, the authors should test other CRBN ligands with alternative scaffolds to assess whether CRBNmidi can respond to these scaffolds in a manner translatable to the full-length, native CRBN. These scaffolds include uracil and other metabolites (PMID: 31459225), phenyl-glutarimides (PMID: 34614283), and cyclimids (PMID: 38320555), all of which have different binding modes from thalidomide analogs and offer a broad range of binding affinities against CRBN.

4. The authors should employ hydrogen-deuterium exchange mass spectrometry (HDX-MS) to gain insights into the dynamics of the CRBNmidi structure, especially in comparison to the previous CRBN-DDB1 construct and in the presence/absence of ligands. While crystallography provides static snapshots, HDX-MS can reveal how parts of CRBNmidi move or change over time. This dynamic information is crucial for understanding the real-time function of CRBNmidi in natural environments. Evaluating this aspect is essential for validating the utility of the new construct, especially given CRBN's known flexibility upon ligand binding and possibly when recognizing native substrates.

Minor points:

- In Figures 1D, 2D, and 2E, some labels have become unreadable due to garbled text.

Below are our responses point-by-point to each of the reviewers' points.

Reviewer #1

I was requested to comment on the SAXS and MD simulations reported in the manuscript "Design of a Cereblon construct for crystallographic and biophysical studies of protein degraders." Overall, the application of SAXS and MD simulations support the conclusions of the study. I have only minor comments:

We thank the reviewer for critically reviewing our manuscript and appreciate the constructive feedback. We have appropriately revised our manuscript and provided our responses below to address each comment.

Minor comments:

1. When describing the rearrangement of the TBD and Lon domains between open and closed state the authors provide a reference to Petzold et al. [1]. Maybe I missed it, however, I did not find an open structure in the given reference. Please clarify.

We thank the reviewer for noticing this mistake. We have removed this specific reference to Petzold paper and added a reference to Nowak 2018, which does describe the open conformation of CRBN.

2. The authors report that in absence of a small-molecule ligand, CRBN^{midi} adopted the closed conformation. This contradicts the findings by Watson et al. that apo-CRBN always adopts an open conformation [2]. Could the authors elaborate on the reasons why apo-CRBN^{midi} adopts the closed conformation within the crystal? Watson et al. suggest that interactions of the sensor loop with a helix in the HB domain (residues ~210 to 220) and with a loop in BPC in DDB1 (residues ~776 to 780) play a role in maintaining the open conformation of CRBN [2]. Could the authors comment on the role of the sensor loop in the apo-CRBN^{midi} crystal?

As Reviewer 1 highlights, the open conformation of CRBN-DDB1 that is observed in Watson et al.'s cryo-EM structure appears to be stabilised through interaction of the sensor loop (residues 341-361) with residues ~776-780 in DDB1. As our CRBN^{midi} construct does not contain DDB1 to enforce the open conformation we were able to access the closed conformation, which was amenable to crystallisation (Fig. 1C). In this closed conformation the sensor loop adopts a near identical conformation in both binder-complexed and apo crystal structures. Ultimately, any construct of CRBN

containing both Lon and TBD will exhibit an equilibrium between open-closed that could be modulated and biased e.g. through crystallization.

We have elaborated on our description of the apo crystal structure to highlight this point.

3. The authors compared their measured SAXS intensities to back-calculated curves from the crystal structure of CRBN^{mid}:mezigdomide in closed conformation and a homology model of the open conformation using CRY SOL. Notably, while using the same model in the first and third graph and second and fourth graph, respectively, the back-calculated curves are different. This is possibly the result of the large number of fitting parameters used by CRY SOL (for the hydration shell, two for the excluded solvent, optionally a constant offset). The authors may consider using a method that uses explicit solvent and thus does not require free fitting parameters for back-calculating a SAXS curve, such as WAXSiS [3,4] or CAPRIQORN [5].

The reviewer is correct in both their observation and the proposed explanation. A comparison is attached where scattering curves were predicted by either WAXSiS using explicit solvent or calculated to fit using CRY SOL to the experimental curves for apo or mezigdomide-bound CRBN^{mid}.

It is obvious that the CRY SOL fits are very similar to the WAXSiS curves for the case that the matching experimental curve is used but leads to significant distortions in case of mismatch. However, this does not affect our conclusions on the solution structure of CRBN^{mid} and the CRY SOL approach is widely accepted in the field as the explicit modelling of the solvent layer can be difficult and can make it hard to obtain good fits even if solution structure and input model are very similar. This was also confirmed by discussion with P. Chen, main author of WAXSiS.

Since this has no impact on the conclusions of our analysis, we feel that it is best to stay with the widely accepted approach used by CRY SOL, but we thank the reviewer for pointing out this interesting observation and encouraging us to explore the most appropriate method for fitting the SAXS curves!

4. In the section on Molecular Dynamics some details are missing:

a. Please provide details how the structure 8D7U was modified, e.g. that DDB1 was removed.

The 'Materials and Methods' section has been revised to mention DDB1 chain was not considered for the M.D. simulations.

b. Please explain over which atoms the RMSF was calculated, such as C α , backbone or heavy atoms

The 'Materials and Methods' section has been revised to mention RMSF was calculated over heavy atoms.

c. Please provide details on how the Protein-ligand interaction stability were calculated.

The protein residues that are involved in various types of non-covalent interactions with the ligand atoms for more than 20% of the entire simulation time (100 ns) in each run were noted to compare the interaction profiles (interaction types and their stabilities) of mezigdomide bound to the two different protein constructs – CRBN^{mid} and CRBN^{AN}. We have now revised the 'Materials and Methods' section to include this explanation on how we have calculated the Protein-ligand interaction stability. Also, the legend to Fig. S3 is revised to address this comment.

d. The authors report on the 9Å cut-off radius that was set for the Coulomb interactions. Plain cut-offs for Coulomb interactions are uncommon since 25 years because they may lead to severe artifacts. Was really a plain cut-off used, and not, for instance, particle-mesh Ewald (PME)? Do the authors possibly refer to the direct-space cutoff within the PME method?

We used the u-series decomposition method to calculate the Coulomb interactions in our M.D. simulation studies as reported by Predescu *et al.* from the D.E. Shaw Research group in the *J. Chem. Phys* (2020). This method decomposes the Coulomb potential into near and far parts that is constructed to be exact up to the cutoff radius (r_c) and continuous at the cutoff. As a result, the magnitude of the Coulomb interaction error near r_c is decreased relative to the Ewald decomposition. The authors have shown that

u-series decomposition of the Coulomb potential is more accurate than the standard (Ewald) decomposition for a given amount of computational effort. Also, it achieves the same accuracy as the Ewald decomposition with approximately half the computational effort.

We have now revised the manuscript to indicate in the 'Materials and Methods' that we have used 'u-series' decomposition method to calculate Coulomb interactions and cited the above-mentioned article.

5. In Figure 1 A there is a square in the superscript, which likely should be a Δ to report on CRBN Δ N.

Thank you for highlighting this, this was an issue that arose during figure export; we have now amended the figures.

6. I suggest adding a reference to figure 1 D when discussing the superposition of CRBN_{mid} with the corresponding portion of PDB ID 4TZ4.

Reference added.

7. Please add units to the RMSF in figure S3 A, B, C, D.

Thank you for highlighting this omission, the axes and legend in Fig. S3 have been revised to include the unit of RMSF, i.e., Å.

8. I suggest adding a line legend to figure 3 or describing the lines in the caption.

Thank you for highlighting this omission, we've added a line legend to the SPR graphs.

[1] G. Petzold, E.S. Fischer & N.H. Thomä, Structural basis of lenalidomide-induced CK1 α degradation by the CRL4CRBN ubiquitin ligase. *Nature* 532, 127-130 (2016).

[2] E.R. Watson et al. Molecular glue CELMoD compounds are regulators of cereblon conformation. *Science* 378, 549-553 (2022)

[3] P. Chen and J.S. Hub, Validating solution ensembles from molecular dynamics simulations by wide-angle X-ray scattering data, *Biophys. J.*, 107, 435-447 (2014)

[4] C.J. Knight, J.S. Hub, WAXSiS: a web server for the calculation of SAXS/WAXS curves based on explicit-solvent molecular dynamics. *Nucleic Acids Res.* 43(W1), W225-W230 (2015)

[5] J. Köfinger, G. Hummer, Atomic-resolution structural information from scattering experiments on macro- molecules in solution. *Phys. Rev. E* 87(5), 052712 (2013)

Reviewer #2 (Remarks to the Author)

Although CRBN has been developed as a major E3 ligase handle for degrader molecules, there is a relative paucity in structural data for ternary complexes with neo-substrates. Some biophysical studies are also impeded by challenges in producing quality protein due to both CRBN's intrinsic conformational heterogeneity and a requirement for coexpression with DDB1. The present work provides a new system for facile production of a "bonsai", or miniaturized version of CRBN. This CRBNmidi retains the two domains that are crucial for binding to degrader molecules and substrates (the Lon domain and thalidomide binding domain, TBD). The authors achieved their goal after using standard protein engineering approaches of deleting the flexible region, domains that are unnecessary for ternary complex formation, introducing solubilizing mutations in place of "sticky" hydrophobic surfaces, and a C/S mutation known to improve crystallization. The culmination of these efforts yielded 1.5 mg/liter of pure, CRBNmidi using straightforward approaches for expression in *E. coli* and purification.

The authors pressure test CRBNmidi for recapitulation of native CRBN features and utility in a battery of assays used for developing degrader molecules. They performed SACS analysis that suggested that degrader-bound CRBNmidi has a smaller radius of gyration than the apo protein. This idea is also supported by MD simulations. They show CRBNmidi performs well in biophysical methods of TR-FRET, SPR, and ITC.

The authors solved crystal structures of CRBNmidi in complexes with lenolidomide and mezigdomide:IKZF1ZF2 that superimpose well with the known structures. They solve a structure of CRBNmidi:mezigdomide that reveal an additional interaction with the ligand fluorine. And they obtained a new structure: CRBNmidi:PROTAC CFT-1297:BRD4BD2. This had two molecules in the asymmetric unit that slightly differ in relative interaction with the neo-substrate.

Overall, the study reports a new reagent, CRBNmidi, that promises substantially reduce the effort and cost for biophysical and structural analysis of ligand and ternary complex interactions with the most predominant E3 ligase used in targeted protein degradation. This will democratize the field by making such studies broadly accessible, and could greatly advance development of degrader reagents and ideally therapeutics. As such, I recommend publication in Nature Communications after addressing the comments below.

We thank the reviewer for critically reviewing our manuscript and appreciate the constructive feedback. We have appropriately revised our manuscript and provided our responses below to address each comment.

Main comments:

The authors' own findings that CRBN^{midi}:PROTAC CFT-1297:BRD4BD2 crystallizes in two orientations highlights an additional potential challenge to obtaining CRBN ternary complex structures: neosubstrate binding could be heterogeneous. Crystallography can aid in obtaining structures of such complexes through crystal packing but such interactions often limit resolution by cryo-EM.

1. This point should be raised in the discussion.

The value of visualising structural flexibility through capturing multiple orientations in different protomers in the ASU has been expanded upon in the discussion of the CRBN^{midi}:CFT-1297:BRD4^{BD2} crystal structure. Further demonstration of this point has also been provided as part of the analysis of the new CRBN^{midi}:compound **1** cocrystal structure.

2. The authors should show the crystal packing interactions to reveal the extent to which interactions in the binary and ternary complexes are stabilized by crystal packing. If the lattice constrains positions available to the neo substrate, this should be described in the results and discussion. The discussion could frame this as both a strength and a weakness.

We thank the reviewer for raising the important issue of crystal packing. We have re-inspected all our crystal structures to assess whether crystal packing affects the position of ligands and neo-substrates, or the overall fold of the protein. CRBN^{midi} has crystallized in different space groups with crystal contacts mediated by different regions whilst maintaining the same fold and ligand/neosubstrate conformation as observed in previously published crystal or cryoEM structures. Whilst in the ternary complexes, there are crystal contacts distributed uniformly across both CRBN^{midi} and the target protein, we have seen no indication that the crystal packing would trigger any artefactual conformations.

We have added a paragraph addressing this in the “High-resolution crystal structures and biophysical characterization of degrader ternary complexes enabled by CRBN^{midi}” results section and have illustrated the ternary crystal packing in the new figure S5.

Reviewer #3 (Remarks to the Author):

Reviewer #4 (Remarks to the Author)

The manuscript presents CRBNmidi, a new cereblon (CRBN) construct engineered for high-yield expression from *E. coli*. The ubiquitin E3 ligase CRBN, utilized extensively in targeted protein degradation, serves as a recruitment platform for PROTACs and molecular glue degraders. However, structure-guided drug design and *in vitro* biophysical characterization of CRBN-based degraders were partially impeded by difficulties in producing a suitable recombinant CRBN. CRBNmidi, which retains the Lon and thalidomide binding domains but excludes the DDB1 binding domain, addresses this issue by enhancing solubility and stability without the adaptor protein DDB1, avoiding the self-aggregation typically observed when CRBN is expressed alone. Through high-resolution X-ray crystallography and cryoEM of CRBNmidi with degrader molecules such as lenalidomide, mezigdomide, and CFT-1297, the authors have elucidated detailed protein-ligand and protein-protein interactions previously unobserved with other CRBN constructs. Additionally, the suitability of CRBNmidi for *in vitro* assays has been confirmed through various biophysical ligand-binding studies, including SAXS, DSF, ITC, TR-FRET, and SPR.

CRBNmidi appears to be a promising platform for high-resolution structural studies and the design of new degraders. This work will likely interest those in the protein degradation and drug discovery communities. However, the utility and applicability of CRBNmidi in the context of CRBN's physiological functions and as a tool to study native interactions with its endogenous substrates remain uncertain.

I would support the publication of this manuscript in Nature Communications if the following points are addressed:

We thank the reviewer for critically reviewing our manuscript and appreciate the constructive feedback. We have appropriately revised our manuscript and provided our responses below to address each comment.

1. CRBNmidi lacks certain domains and includes 12 site mutations, so its interaction dynamics might differ from the full-length native CRBN, potentially leading to overlooking some physiological interactions and allosteric effects. The authors should assess the physiological relevance of CRBNmidi by investigating whether it adequately models native CRBN functions. The authors should verify whether CRBNmidi maintains a consistent binding mode with at least one or two known native CRBN substrates (e.g., MEIS2, APP, GLUL) and/or engineered/endogenous proteins with C-terminal cyclic imide degrons.

As highlighted by reviewer 4, C-terminal cyclic imides have recently been identified as

natural degrons of CRBN. To investigate whether our CRBN^{midi} construct was able to recognise this moiety, we synthesised dipeptide C-terminal cyclic imides Boc-VcN and Boc-AcQ. Boc-VcN and Boc-AcQ significantly stabilised CRBN^{midi}, increasing the denaturation temperature of CRBN^{midi} by 11.1°C and 8.3 °C, respectively as determined by DSF. In SAXS experiments, Boc-VcN and Boc-AcQ were also shown to significantly decrease the R_G of CRBN^{midi} in solution from 26.8 Å to 23.4 Å and 24.3 Å, respectively, indicating stabilisation of the closed CRBN^{midi} conformation on ligand binding. Together these experiments validate the capacity of CRBN^{midi} to bind to di-peptidic C-terminal cyclic imide motifs as representative epitopes of the native degron of native CRBN. These data have been added to Figure 2.

2. The manuscript's claim that CRBN^{midi}'s readiness to crystallize in a ligand-bound form is an improvement over previous constructs should be carefully rephrased to acknowledge its limitations. Specifically, the limited utility of CRBN^{midi} for designing allosteric binders of CRBN or small molecules that influence the open vs. closed conformation of CRBN upon binding should be clearly stated.

We have added an acknowledgement of this caveat to the discussion section.

3. The engineered mutations and deletions might artificially stabilize specific conformations, as evidenced by apo CRBN^{midi} adopting a closed conformation without ligands, which could bias structural and functional outcomes and may render CRBN^{midi} unresponsive to binders with weaker affinities than well-characterized thalidomide derivatives. Since molecular glues with weaker binding affinities are beneficial due to their minimal inhibitory effects on E3 ligase interactions with native substrates, the authors should test other CRBN ligands with alternative scaffolds to assess whether CRBN^{midi} can respond to these scaffolds in a manner translatable to the full-length, native CRBN. These scaffolds include uracil and other metabolites (PMID: 31459225), phenyl-glutarimides (PMID: 34614283), and cyclimids (PMID: 38320555), all of which have different binding modes from thalidomide analogs and offer a broad range of binding affinities against CRBN.

In order to demonstrate that CRBN^{midi} has the potential to illuminate binding modes of non-imide-based binders, we solved the novel crystal structure of CRBN^{midi} in complex with compound **1**, a phenyl glutarimide-based binder, to 1.95 Å. To our knowledge, this is the first reported crystal structure of human CRBN in complex with a phenyl-glutarimide binder. We have added details of this complex to Figure 2.

4. The authors should employ hydrogen-deuterium exchange mass spectrometry (HDX-MS) to gain insights into the dynamics of the CRBN^{midi} structure, especially in

comparison to the previous CRBN-DDB1 construct and in the presence/absence of ligands. While crystallography provides static snapshots, HDX-MS can reveal how parts of CRBN^{midi} move or change over time. This dynamic information is crucial for understanding the real-time function of CRBN^{midi} in natural environments. Evaluating this aspect is essential for validating the utility of the new construct, especially given CRBN's known flexibility upon ligand binding and possibly when recognizing native substrates.

Unfortunately, obtaining new HDX-MS data is not currently possible with the funding, resources and expertise available to this project without significantly delaying the publication of this article.

While we agree with reviewer 4's observation that crystallography only provides a snapshot of a protein complex, we feel that the data package that we have presented convincingly delivers the goals of our project: to demonstrate the utility of CRBN^{midi} as a structural and biophysical tool for the development of CRBN-recruiting degraders. We feel that comparing the relative dynamics and open-close equilibria of various constructs of CRBN, including CRBN^{midi} would be best explored in a future publication.

Minor points:

- In Figures 1D, 2D, and 2E, some labels have become unreadable due to garbled text. We thank the reviewer for flagging this issue. We have replaced the figures in question with updated versions that correctly display the intended symbols.

REVIEWERS' COMMENTS

Reviewer #1 (Remarks to the Author):

My technical comments have been adequately addressed.

As a final note, I don't agree with the assumption that finding "a good fit" between an experimental SAXS curve and a theoretically computed curve by simply adding more fitting parameters (as CRYSOLO does) has any value. Adding more fitting parameters always lowers χ^2 ; this is what a fit does, but it does not provide greater confidence in a structural model. More fitting parameters typically just absorb errors in the structural model.

However, I agree with the reviewers that, for the present application, this discussion does not affect the main conclusions drawn from the SAXS data. Therefore, for future projects, I merely encourage the authors to consider using other methods with fewer fitting parameters (such as Capricorn or Waxis).

Reviewer #2 (Remarks to the Author):

In their revised manuscript, the authors have added new data, new analyses, and new text that collectively address reviewer comments and strengthen the manuscript. The new structure is particularly a welcome addition. I support publication of this work in Nature Communications.

Reviewer #3 (Remarks to the Author):

Reviewer #4 (Remarks to the Author):

The authors have addressed the majority of my concerns from the first review. They have demonstrated that the CRBN_{mid} construct can recognize not only thalidomide-based structures but also non-imide-based binders, such as phenylglutarimide, which possess lower binding affinity to CRBN, as well as an epitope of the native CRBN degron. This solidifies the construct's utility and robustness. Additionally, the newly added text on page 6, which elaborates on a possible reason for observing a closed conformation in their structural data in the absence of a small-molecule ligand, provides readers with valuable insights into the CRBN_{mid}'s behavior during crystallization and the factors influencing its structural state. I support the publication and would like to congratulate the authors on their elegant work.

Author's Response to the Reviewer #1 Final Comments:

As a final note, I don't agree with the assumption that finding "a good fit" between an experimental SAXS curve and a theoretically computed curve by simply adding more fitting parameters (as CRY SOL does) has any value. Adding more fitting parameters always lowers χ^2 ; this is what a fit does, but it does not provide greater confidence in a structural model. More fitting parameters typically just absorb errors in the structural model.

However, I agree with the reviewers that, for the present application, this discussion does not affect the main conclusions drawn from the SAXS data. Therefore, for future projects, I merely encourage the authors to consider using other methods with fewer fitting parameters (such as Capricorn or Waxisis).

We thank the reviewer for this suggestion. We are aware of the discussion in the field about the impact of solvation layers and the problem of overfitting and we would be keen to explore this in future with data that lends itself to a more quantitative analysis. However, we believe that the implicit treatment of the solvation layer has advantages for the purely qualitative analysis we report here, as it avoids potential bias from explicit modelling of the solvation layer, e.g., through the impact of organic co-solvents. Because of this limitation, and as the limited information content of SAXS curves usually do not allow for an unambiguous identification of a model, throughout the manuscript we describe the models as consistent or inconsistent with experimental curves, rather than describing them as "good fits", and focus the analysis of the SAXS data on radius of gyration and Kratky plot, that are obtained without fitting.